# The Good, the Bad and the Ugly: Watermarks, Transferable Attacks and Adversarial Defenses

## Abstract

We formalize and extend existing definitions of backdoor-based watermarks and adversarial defenses as *interactive protocols* between two players. The existence of these schemes is inherently tied to the learning tasks for which they are designed. Our main result shows that for *almost every* learning task, at least one of the two – a watermark or an adversarial defense – exists. The term "almost every" indicates that we also identify a third, counterintuitive but necessary option, i.e., a scheme we call a *transferable attack*. By transferable attack, we refer to an efficient algorithm computing queries that look indistinguishable from the data distribution and fool *all* efficient defenders. To this end, we prove the necessity of a transferable attack via a construction that uses a cryptographic tool called homomorphic encryption. Furthermore, we show that any task that satisfies our notion of a transferable attack implies a *cryptographic primitive*, thus requiring the underlying task to be computationally complex. These two facts imply an "*equivalence*" between the existence of transferable attacks and cryptography. Finally, we show that the class of tasks of bounded VC-dimension has an adversarial defense, and a subclass of them has a watermark.

## 1 Introduction

A company invested considerable resources to train a new classifier $f$. They want to open-source $f$ but also ensure that if someone uses $f$, it can be detected in a black-box manner. In other words, they want to embed a *watermark* into $f$.[1] Alice, an employee, is in charge of this project. Bob, a member of an AI Safety team, has a different task. His goal is to make $f$ *adversarially robust*, i.e., to ensure it is hard to find queries that appear unsuspicious but cause $f$ to make mistakes. Alice, after many unsuccessful approaches, reports to her boss that it might be inherently impossible to create a black-box watermark in $f$ that cannot be removed. After a similar experience, Bob reports to his boss that, due to the sheer number of possible modes of attack, he was only able to produce an ever-growing, 'ugly' defense.

One day, after discussing their respective projects, Alice and Bob realized that their projects are intimately connected. Alice said that her idea was to plant a backdoor in $f$, creating $f_A$, so she could later craft queries with a *hidden trigger* that activates the backdoor, causing $f_A$ to misclassify, while remaining *indistinguishable* from standard queries. By sending these tailored queries in a black-box manner to a party suspected of using $f_A$, she can detect whether $f_A$ is being used based on the responses triggered by her backdoor. But Bob realized that his defenses were trying to render such a situation impossible. One of his ideas for defense was to take $f$ and then "smooth" its outputs to obtain $f_B$, aiming for robustness against attacks. Bob noticed that this procedure removes some of the backdoor-based watermarks that Alice came up with. Conversely, Alice noticed that any $f$ with a watermark that is difficult to remove implies that some models are inherently difficult to make robust. Alice and Bob realized that their challenges are two sides of the same coin: the impossibility of one task guarantees the success of the other.

---

[1]Note that they want to watermark the model itself, not its outputs.

## 1.1 CONTRIBUTIONS

This paper initiates a formal study of the above observation that backdoor-based watermarks and adversarial defenses span all possible scenarios. By scenarios, we refer to learning tasks that $f$ is supposed to solve.

*Our main contribution is:*

> *We prove that almost every learning task has at least one of the two:*
> *A Watermark or an Adversarial Defense.*

To do that, we formalize and extend existing definitions of watermarks and adversarial defenses, frame Alice and Bob's dynamic as a formal game, and show that this game is guaranteed to have at least one winner. Along the way to proving the main result, we identify a potential reason why this fact was not discovered earlier. There is also a third, counterintuitive but necessary option, i.e., *there are tasks with neither a Watermark nor an Adversarial Defense*.

Imagine that Alice plays the following game. The game is played with respect to a specific learning task $\mathcal{L} = (\mathcal{D}, h)$, where $\mathcal{D}$ is the data distribution and $h$ is the ground truth. Alice sends queries to a player and receives their responses. She wins if the responses have a lot of errors and if the player cannot distinguish them from the queries from $\mathcal{D}$. Importantly, whether she wins the game depends on how much compute and data Alice and the player have. If Alice wins the game against any player having the same amount of resources as her, then we call Alice's queries a *Transferable Attack*. Intuitively, the harder a query becomes, the easier it is to distinguish it from queries from $\mathcal{D}$. But this seems to indicate that it is hard to design Transferable Attacks.

However, we provably show:

- An example of a Transferable Attack defined as above. Interestingly, the example uses tools from the field of cryptography, namely Fully Homomorphic Encryption (FHE) (Gentry, 2009). Notably, a Transferable Attack rules out Watermarks and Adversarial Defenses, thus constituting the third necessary option.

- That every Transferable Attack implies a certain *cryptographic primitive*, i.e., access to samples from the underlying task is enough to build essential parts of encryption systems. Thus, every task with a Transferable Attack has to be complex in the computational complexity theory sense.

Finally, we complement the above results with instantiations of Watermarks and Adversarial Defenses:

- We show the existence of an Adversarial Defense for all learning tasks with bounded Vapnik–Chervonenkis (VC) dimension, thereby ruling out Transferable Attacks in this regime.

- We give an example of a black-box Watermark for a class of learning tasks with bounded VC-dimension. Notably, in this case, both a Watermark and an Adversarial Defense exist.

## 2 RELATED WORK

This paper lies at the intersection of machine learning theory, interactive proof systems, and cryptography. We review recent advances and related contributions from these areas that closely align with our research.

**Interactive Proof Systems in Machine Learning.** *Interactive Proof Systems* (Goldwasser & Sipser, 1986) have recently gained considerable attention in machine learning for their ability to formalize and verify complex interactions between agents, models, or even human participants. A key advancement in this area is the introduction of *Prover-Verifier Games* (PVGs) (Anil et al., 2021), which employ a game-theoretic approach to guide learning agents towards decision-making with verifiable outcomes. Building on PVGs, Kirchner et al. (2024) enhance this framework to improve the legibility of Large Language Models (LLMs) outputs, making them more accessible for human evaluation. Similarly, Wäldchen et al. (2024) apply the prover-verifier setup to offer interpretability guarantees for classifiers.

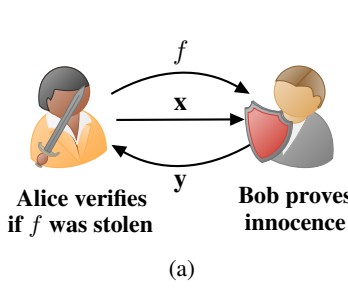

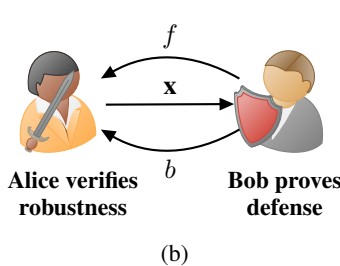

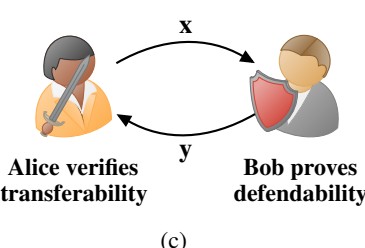

| (a) | (b) | (c) |
|---|---|---|

A **Watermark** is an efficient algorithm that computes a low-error classifier $f$ and a set of queries $\mathbf{x}$ such that (fast) defenders are unable to find low-error answers $\mathbf{y}$ nor distinguish $\mathbf{x}$ from the data distribution.

An **Adversarial Defense** is an efficient algorithm that computes a low-error classifier $f$ and a detection bit $b$, such that (fast) adversaries are unable to find queries $\mathbf{x}$, which look indistinguishable from the data distribution and where $f$ is incorrect.

A **Transferable Attack** is an efficient algorithm that computes queries $\mathbf{x}$ that look indistinguishable from the data distribution, and that fool all efficient defenders.

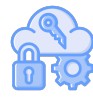

Figure 1: Schematic overview of the interaction structure, along with short, informal versions of our definitions of (a) Watermark (Definition 1), (b) Adversarial Defense (Definition 2), and (c) Transferable Attack (Definition 3), with (c) tied to cryptography (see Section 5).

Extending these concepts, self-proving models Amit et al. (2024) introduce generative models that not only produce outputs but also generate proof transcripts to validate their correctness. In the context of AI safety, scalable *debate protocols* (Condon et al., 1993; Irving et al., 2018; Brown-Cohen et al., 2023) leverage interactive proof systems to enable complex decision processes to be broken down into verifiable components, ensuring reliability even under adversarial conditions.

Overall, these developments highlight the emerging role of interactive proof systems in addressing key aspects of AI Safety, such as interpretability, verifiability, and alignment. While current research predominantly focuses on applying this framework to improve these safety attributes, our approach takes an orthogonal direction by examining the *feasibility* of properties related to *adversarial robustness* and *backdoor-based watermarks*.

**Planting Undetectable Backdoors.** A key related work is presented by Goldwasser et al. (2022), which demonstrates how a learner can plant undetectable backdoors in any classifier, allowing hidden manipulation of the model's output with minimal perturbation of the input. These backdoors are activated by specific *"triggers"*, which are subtle changes to the input that cause the model to misclassify *any* input with the trigger applied, while maintaining its expected behavior on regular inputs. The authors propose two frameworks. The first utilizes digital signature schemes (Goldwasser et al., 1985) that make backdoored models indistinguishable from the original model to any computationally-bounded observer. The second involves Random Fourier Features (RFF) (Rahimi & Recht, 2007), which ensures undetectability even with full transparency of the model's weights and training data.

In a concurrent and independent work, Christiano et al. (2024) introduce a defendability framework that formalizes the interaction between an attacker planting a backdoor and a defender tasked with detecting it. The attacker modifies a classifier to alter its behavior on a trigger input while leaving other inputs unaffected. The defender then attempts to identify this trigger during evaluation, and if successful with high probability, the function class is considered defendable. The authors show an equivalence between their notion of defendability (in a computationally unbounded setting) and Probably Approximately Correct (PAC) learnability, and thus the boundedness of the VC-dimension of a class. In computationally bounded cases, they propose that *efficient defendability* serves as an important intermediate concept between efficient learnability and obfuscation. A major difference between our work and that of Christiano et al. (2024), is that in their approach, the attacker chooses the distribution, whereas we keep the distribution fixed. This makes defendability in their model harder since the attacker has more control. However, in their framework, the backdoor trigger $x^*$

is sampled $\sim \mathcal{D}$, so the attacker does not influence it. In contrast, our model allows the attacker to choose specific $x$'s, making defendability easier in this regard. Thus, the definitions are a priori incomparable. A second major difference is that our main result holds for *all* learning tasks, while their contributions hold only for restricted classes. This makes defendability in their model harder since the attacker has more control. However, in their framework, the backdoor trigger $x^*$ is sampled $\sim \mathcal{D}$, so the attacker does not influence it. In contrast, our model allows the attacker to choose specific $x$'s, making defendability easier in this regard. Thus, the definitions are a priori incomparable. However, there are many interesting connections. Computationally unbounded defendability is shown to be equivalent to PAC learnability, while we, in a similar spirit, show an Adversarial Defense for all tasks with bounded VC-dimension. They show that efficient PAC learnability implies efficient defendability, and we show that the same fact implies an efficient Adversarial Defense. Using cryptographic tools, they show that the class of polynomial-size circuits is not efficiently defendable, while we use different cryptographic tools to give a Transferable Attack, which rules out a Defense.

**Backdoor-Based Watermarks.** In black-box settings, where model auditors lack access to internal parameters, watermarking methods often involve embedding backdoors during training. Techniques by Adi et al. (2018) and Zhang et al. (2018) use crafted input patterns as triggers linked to specific outputs, enabling ownership verification by querying the model with these specific inputs. Advanced methods by Merrer et al. (2017) utilize adversarial examples, which are perturbed inputs that yield predefined outputs. Further enhancements by Namba & Sakuma (2019) focus on the robustness of watermarks, ensuring the watermark remains detectable despite model alterations or attacks.

In the domain of Natural Language Processing (NLP), backdoor-based watermarks have been studied for Pre-trained Language Models (PLMs), as exemplified by works such as (Gu et al., 2022; Peng et al., 2023) and (Li et al., 2023). These approaches embed backdoors using rare or common word triggers, ensuring watermark robustness across downstream tasks and resistance to removal techniques like fine-tuning or pruning. However, it is important to note that these lines of research are predominantly empirical, with limited theoretical exploration.

**Adversarial Robustness.** As we emphasize, the study of backdoors is closely related to adversarial robustness, which focuses on improving model resilience to adversarial inputs. The extensive literature in this field includes key contributions such as *adversarial training* (Madry et al., 2018), which improves robustness by training on adversarial examples, and certified defenses (Raghunathan et al., 2018), which offer *provable guarantees* against adversarial attacks by ensuring prediction stability within specified perturbation bounds. Techniques like *randomized smoothing* (Cohen et al., 2019) extend these robustness guarantees. Notably, Goldwasser et al. (2022) show that some undetectable backdoors can, in fact, be removed by randomized smoothing, highlighting the intersection of adversarial robustness and backdoor methods.

## 3 WATERMARKS, ADVERSARIAL DEFENSES AND TRANSFERABLE ATTACKS

In this section, we outline interactive protocols between a verifier and a prover. Each protocol is designed to address specific tasks such as watermarking, adversarial defense, and transferable attacks. We first introduce the preliminaries before detailing the properties that each protocol must satisfy.

### 3.1 PRELIMINARIES

**Discriminative Learning Task.** For $n \in \mathbb{N}$, we define $[n] := \{0, 1, \ldots, n-1\}$. A *learning task* $\mathcal{L}$ is a pair $(\mathcal{D}, h)$ of a distribution $\mathcal{D}$, $\mathrm{supp}(\mathcal{D}) \subseteq \mathcal{X}$ (the input space), and a ground truth map $h \colon \mathcal{X} \to \mathcal{Y} \cup \{\bot\}$, where $\mathcal{Y}$ is a finite space of labels and $\bot$ represents a situation where $h$ is not defined. To every $f \colon \mathcal{X} \to \mathcal{Y}$, we associate $\mathrm{err}(f) := \mathbb{E}_{x \sim \mathcal{D}}[f(x) \neq h(x)]$. We implicitly assume $h$ does not map to $\bot$ on $\mathrm{supp}(\mathcal{D})$. This definition of $\bot$ is introduced for generality, as it becomes relevant in adversarial scenarios where samples may lie outside $\mathrm{supp}(\mathcal{D})$.

For $q \in \mathbb{N}, \mathbf{x} \in \mathcal{X}^q, \mathbf{y} \in \mathcal{Y}^q$, we define

$$\mathrm{err}(\mathbf{x}, \mathbf{y}) := \frac{1}{q} \sum_{i \in [q]} \mathbb{1}_{\{h(x_i) \neq y_i, h(x_i) \neq \bot\}},$$

which means that we count $(x, y) \in \mathcal{X} \times \mathcal{Y}$ as an error if $h$ is well-defined on $x$ and $h(x) \neq y$.

**Advantage and Indistinguishability:** For an algorithm $\mathcal{A}$ (also known as the distinguisher) and two distributions $\mathcal{D}_0, \mathcal{D}_1$, consider the following game between a sender and the distinguisher:

1. The sender samples a bit $b \sim U(\{0, 1\})$ and then draws a random sample $\mathbf{x} \sim \mathcal{D}_b$.

2. $\mathcal{A}$ receives $\mathbf{x}$ and outputs $\hat{b} := \mathcal{A}(\mathbf{x}) \in \{0, 1\}$. $\mathcal{A}$ wins if $\hat{b} = b$.

We say that $\delta \in \left(0, \frac{1}{2}\right)$ is the *advantage* of $\mathcal{A}$ for *distinguishing* $\mathcal{D}_0$ from $\mathcal{D}_1$ if: $P_{b \sim U(\{0,1\}), \mathbf{x} \sim \mathcal{D}_b}[\mathcal{A}(\mathbf{x}) = b] = \frac{1}{2} + \delta$. For a class of algorithms, we say that the two distributions $\mathcal{D}_0$ and $\mathcal{D}_1$ are $\delta$-*indistinguishable* if for any algorithm in the class, its advantage is at most $\delta$.

### 3.2 DEFINITIONS

In our protocols, Alice (**A**, verifier) and Bob (**B**, prover) engage in interactive communication, with distinct roles depending on the specific task. Each protocol is defined with respect to a learning task $\mathcal{L} = (\mathcal{D}, h)$, an error parameter $\varepsilon \in \left(0, \frac{1}{2}\right)$, and time bounds $T_\mathbf{A}$ and $T_\mathbf{B}$. A scheme is successful if the corresponding algorithm satisfies the desired properties with high probability, and we denote the set of such algorithms by SCHEME($\mathcal{L}, \varepsilon, T_\mathbf{A}, T_\mathbf{B}$), where SCHEME refers to WATERMARK, DEFENSE, or TRANSFATTACK.

**Definition 1** (*Watermark, informal*).
An algorithm $\mathbf{A}_{\text{WATERMARK}}$, running in time $T_\mathbf{A}$, implements a *watermarking scheme* for the learning task $\mathcal{L}$ with error parameter $\epsilon > 0$, if an interactive protocol in which $\mathbf{A}_{\text{WATERMARK}}$ computes a classifier $f : \mathcal{X} \to \mathcal{Y}$ and a sequence of queries $\mathbf{x} \in \mathcal{X}^q$, and a prover $\mathbf{B}$ outputs $\mathbf{y} = \mathbf{B}(f, \mathbf{x}) \in \mathcal{Y}^q$, satisfies the following properties:

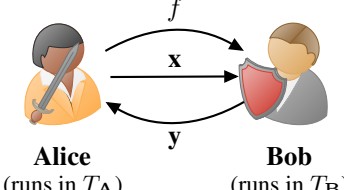

**Alice**
(runs in $T_\mathbf{A}$)

**Bob**
(runs in $T_\mathbf{B}$)

1. **Correctness:** $f$ has low error, i.e., $\text{err}(f) \leq \epsilon$.

2. **Uniqueness:** There exists a prover $\mathbf{B}$, running in time bounded by $T_\mathbf{A}$, which provides low-error answers, such that $\text{err}(\mathbf{x}, \mathbf{y}) \leq 2\epsilon$.

Figure 2: Schematic overview of the interaction between Alice and Bob in *Watermark* (Definition 1).

3. **Unremovability:** For every prover $\mathbf{B}$ running in time $T_\mathbf{B}$, it holds that $\text{err}(\mathbf{x}, \mathbf{y}) > 2\epsilon$.

4. **Undetectability:** For every prover $\mathbf{B}$ running in time $T_\mathbf{B}$, the advantage of $\mathbf{B}$ in distinguishing the queries $\mathbf{x}$ generated by $\mathbf{A}_{\text{WATERMARK}}$ from random queries sampled from $\mathcal{D}^q$ is small.

Note that, due to *uniqueness*, we require that any defender, who *did not use $f$* and trained a model $f_{\text{Scratch}}$, must be accepted as a distinct model. This requirement is essential, as it mirrors real-world scenarios where independent models could have been trained within the given time constraint $T_\mathbf{A}$. Additionally, the property enforces that any successful Watermark must satisfy the condition that Bob's time is strictly less than $T_\mathbf{A}$, i.e., $T_\mathbf{B} < T_\mathbf{A}$.

**Definition 2** (*Adversarial Defense, informal*).
An algorithm $\mathbf{B}_{\text{DEFENSE}}$, running in time $T_\mathbf{B}$, implements an *adversarial defense* for the learning task $\mathcal{L}$ with error parameter $\epsilon > 0$, if an interactive protocol in which $\mathbf{B}_{\text{DEFENSE}}$ computes a classifier $f : \mathcal{X} \to \mathcal{Y}$, a verifier $\mathbf{A}$ replies with $\mathbf{x} = \mathbf{A}(f)$, where $\mathbf{x} \in \mathcal{X}^q$, and $\mathbf{B}_{\text{DEFENSE}}$ outputs $b = \mathbf{B}_{\text{DEFENSE}}(f, \mathbf{x}) \in \{0, 1\}$, satisfies the following properties:

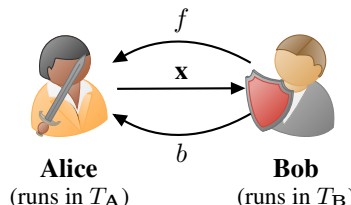

**Alice**
(runs in $T_\mathbf{A}$)

**Bob**
(runs in $T_\mathbf{B}$)

1. **Correctness:** $f$ has low error, i.e., $\text{err}(f) \leq \epsilon$.

2. **Completeness:** When $\mathbf{x} \sim \mathcal{D}^q$, then $b = 0$.

3. **Soundness:** For every $\mathbf{A}$ running in time $T_\mathbf{A}$, we have $\text{err}(\mathbf{x}, f(\mathbf{x})) \leq 7\epsilon$ or $b = 1$.

Figure 3: Schematic overview of the interaction between Alice and Bob in *Adversarial Defense* (Definition 2).

The key requirement for a successful defense is the ability to *detect when it is being tested*. To bypass the defense, an attacker must provide samples that are both *adversarial*, causing the classifier to make mistakes, and *indistinguishable* from samples drawn from the data distribution $\mathcal{D}$.

**Definition 3** (*Transferable Attack, informal*).

An algorithm $\mathbf{A}_{\text{TRANSFATTACK}}$, running in time $T_{\mathbf{A}}$, implements a *transferable attack* for the learning task $\mathcal{L}$ with error parameter $\epsilon > 0$, if an interactive protocol in which $\mathbf{A}_{\text{TRANSFATTACK}}$ computes $\mathbf{x} \in \mathcal{X}^q$ and $\mathbf{B}$ outputs $\mathbf{y} = \mathbf{B}(\mathbf{x}) \in \mathcal{Y}^q$ satisfies the following properties:

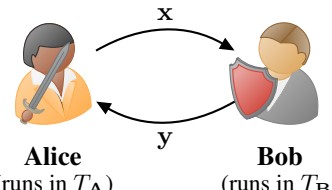

**Alice**
(runs in $T_{\mathbf{A}}$)

**Bob**
(runs in $T_{\mathbf{B}}$)

1. **Transferability:** For every prover $\mathbf{B}$ running in time $T_{\mathbf{A}}$, we have $\text{err}(\mathbf{x}, \mathbf{y}) > 2\epsilon$.

2. **Undetectability:** For every prover $\mathbf{B}$ running in time $T_{\mathbf{B}}$, the advantage of $\mathbf{B}$ in distinguishing the queries $\mathbf{x}$ generated by $\mathbf{A}_{\text{TRANSFATTACK}}$ from random queries sampled from $\mathcal{D}^q$ is small.

Figure 4: Schematic overview of the interaction between Alice and Bob in *Adversarial Defense* (Definition 3).

**Verifiability of Watermarks.** For a watermarking scheme $\mathbf{A}_{\text{WATERMARK}}$, if the *unremovability* property holds with a stronger guarantee, i.e., much larger than $2\epsilon$, then $\mathbf{A}_{\text{WATERMARK}}$ could determine whether $\mathbf{B}$ had stolen $f$. To achieve this, $\mathbf{A}_{\text{WATERMARK}}$ runs, after completing its interaction with $\mathbf{B}$, the procedure guaranteed by *uniqueness* to obtain $\mathbf{y}'$. It then verifies whether $\mathbf{y}$ and $\mathbf{y}'$ differ for many queries. If this condition is met, $\mathbf{A}_{\text{WATERMARK}}$ concludes that $\mathbf{B}$ had stolen $f$.[2] Alternatively, if *unremovability* holds with $2\epsilon$, as originally defined, the test described above may fail. In this scenario, we consider an external party overseeing the interaction, potentially with knowledge of the distribution and $h$, who can directly compute the necessary errors to make a final decision. This setup is similar to the use of human judgment oracles in (Brown-Cohen et al., 2023). An interesting direction for future work would be to explore cases where the parties have access to *restricted* versions of error oracles. While this is beyond the scope of this work, we outline potential avenues for addressing this in Appendix E.

## 4 MAIN RESULT

We are ready to state an informal version of our main theorem. Please refer to Theorem 5 for the details and full proof. The key idea is to define a *zero-sum game* between $\mathbf{A}$ and $\mathbf{B}$, where the actions of each player are the possible algorithms or circuits that can be implemented in the given time bound. Zero-sum games are not a modeling choice but a proof strategy, as they allow us to analyze the complementary nature of attacks on watermarks and adversarial defenses with clean mathematical guarantees. Specifically, the unique value of a zero-sum game eliminates concerns about equilibrium selection. Notably, this game is finite, but there are exponentially many such actions for each player. We rely on some key properties of such large zero-sum games (Lipton & Young, 1994b; Lipton et al., 2003) to argue about our main result. The formal statement and proof is deferred to Appendix D.

**Theorem 1** (*Main Theorem, informal*). *For every learning task $\mathcal{L}$ and $\epsilon \in \left(0, \frac{1}{2}\right), T \in \mathbb{N}$, where a learner exists that runs in time $T$ and, with high probability, learns $f$ satisfying $\text{err}(f) \leq \epsilon$, at least one of these three exists:*

$$\text{WATERMARK}\left(\mathcal{L}, \epsilon, T, T^{1/\sqrt{\log(T)}}\right),$$

$$\text{DEFENSE}\left(\mathcal{L}, \epsilon, T^{1/\sqrt{\log(T)}}, O(T)\right),$$

$$\text{TRANSFATTACK}\left(\mathcal{L}, \epsilon, T, T\right).$$

*Proof (Sketch).* The intuition of the proof relies on the complementary nature of Definitions 1 and 2. Specifically, every attempt to remove a fixed Watermark can be transformed to a potential Adversarial

---

[2]Observe that this test *would not work*, if there were many valid labels for a given input, i.e., a situation often encountered in large language models.

Defense, and vice versa. We define a zero-sum game $\mathcal{G}$ between watermarking algorithms $\mathbf{A}$ and algorithms attempting to remove a watermark $\mathbf{B}$. The use of a zero-sum game ensures that the value of the game is unique, allowing us to focus on the interplay between watermarking and adversarial defenses without ambiguity about equilibrium selection. The actions of each player are the class of algorithms that they can run in their respective time bounds, and the payoff is determined by the probability that the errors and rejections meet specific requirements. According to Nash's theorem, there exists a Nash equilibrium for this game, characterized by strategies $\mathbf{A}_{\text{NASH}}$ and $\mathbf{B}_{\text{NASH}}$. This equilibrium framework simplifies the analysis since Nash equilibria are well-studied and provide tractable guarantees for two-player zero-sum games.

A careful analysis shows that depending on the value of the game, we have a Watermark, an Adversarial Defense, or a Transferable Attack. In the first case, where the expected payoff at the Nash equilibrium is greater than a threshold, we show there is an Adversarial Defense. We define $\mathbf{B}_{\text{DEFENSE}}$ as follows. $\mathbf{B}_{\text{DEFENSE}}$ first learns a low-error classifier $f$, then sends $f$ to the party that is attacking the Defense, then receives queries $\mathbf{x}$, and simulates $(\mathbf{y}, b) = \mathbf{B}_{\text{NASH}}(f, \mathbf{x})$. The bit $b = 1$ if $\mathbf{B}_{\text{NASH}}$ thinks it is attacked. Finally, $\mathbf{B}_{\text{DEFENSE}}$ replies with $b' = 1$ if $b = 1$, and if $b = 0$ it replies with $b' = 1$ if the fraction of queries on which $f(\mathbf{x})$ and $\mathbf{y}$ differ is high. Careful analysis shows $\mathbf{B}_{\text{DEFENSE}}$ is an Adversarial Defense. In the second case, where the expected payoff at the Nash equilibrium is below the threshold, we have either a Watermark or a Transferable Attack. The reason that there are two cases is due to the details of the definition of $\mathcal{G}$. Full proof can be found in Appendix D. $\qquad\square$

Our Definitions 1, 2, 3 and Theorem 1 are phrased with respect to a *fixed* learning task, while VC-theory takes an alternate viewpoint that tries to show guarantees on the risk (mostly sample complexity-based) for any distribution. However, for DNNs and other modern architectures, moving beyond classical VC-theory is necessary (Zhang et al., 2021; Nagarajan & Kolter, 2019). In our case, due to the requirements of our schemes (e.g., *unremovability* and *undetectability*), it may not be feasible to achieve a formalization that applies to all distributions, as in classical VC-theory. We end this section with the following observation.

**Fact 1** (*Transferable Attacks are disjoint from Watermarks and Adversarial Defenses*)**.** For every learning task $\mathcal{L}$ and $\epsilon \in \left(0, \frac{1}{2}\right), T \in \mathbb{N}$, if $\textsc{TransfAttack}\left(\mathcal{L}, \epsilon, T, T\right)$ exists, then neither $\textsc{Watermark}\left(\mathcal{L}, \epsilon, T, o(T)\right)$ nor $\textsc{Defense}\left(\mathcal{L}, \epsilon, T, T\right)$ exists.

This result follows straightforwardly from rephrasing the Definitions 1 to 3. Indeed, a Transferable Attack is a strong notion of an attack, so it rules out a Defense. Secondly, a Transferable Attack against defenders running in time $T$ rules out a Watermark, since it is in conflict with *uniqueness*.

## 5 Transferable Attacks are "equivalent" to Cryptography

In this section, we show that tasks with Transferable Attacks exist. To construct such examples, we use cryptographic tools. But importantly, the fact that we use cryptography is not coincidental. As a second result of this section, we show that every learning task with a Transferable Attack *implies* a certain cryptographic primitive. One can interpret this as showing that Transferable Attacks exist only for *complex learning tasks*, in the sense of computational complexity theory. The two results together justify, why we can view Transferable Attacks and the existence of cryptography as "equivalent".

### 5.1 A Cryptography-based Task with a Transferable Attack

Next, we give an example of a cryptography-based learning task with a Transferable Attack. The following is an informal statement of the first theorem of this section. The formal version (Theorem 7) is given in Appendix G.

**Theorem 2** (*Transferable Attack for a Cryptography-based Learning Task, informal*)**.** *There exists a learning task $\mathcal{L}^{crypto}$ with a distribution $\mathcal{D}$ and hypothesis class $\mathcal{H}$, and $\mathbf{A}$ such that for all $\epsilon$ if $h$ is sampled from $\mathcal{H}$ then*

$$\mathbf{A} \in \textsc{TransfAttack}\left((\mathcal{D}, h), \epsilon, T_{\mathbf{A}} \approx \frac{1}{\epsilon}, T_{\mathbf{B}} = \frac{1}{\epsilon^2}\right).$$

*Moreover, the learning task is such that for every $\epsilon$, $\approx \frac{1}{\epsilon}$ time (and $\approx \frac{1}{\epsilon}$ samples) is enough, and $\approx \frac{1}{\epsilon}$ samples (and in particular time) is necessary to learn a classifier of error $\epsilon$.*

Notably, the parameters are set so that **A** (the party computing **x**) has *less* time than **B** (the party computing **y**), specifically $\approx 1/\epsilon$ compared to $1/\epsilon^2$. Furthermore, because of the encryption scheme, this is a setting where a single input maps to multiple outputs, which deviates away from the setting of classification learning tasks considered in Theorem 1.

*Proof (Sketch).* We start with a definition of a learning task that will be later augmented with a cryptographic tool to produce $\mathcal{L}^{\text{crypto}}$.

**Lines on Circle Learning Task $\mathcal{L}^\circ$ (Figure 5).** Consider a binary classification task $\mathcal{L}^\circ$, where the input space is defined as $\mathcal{X} = \{x \in \mathbb{R}^2 \mid \|x\|_2 = 1\}$, representing points on the unit circle. The hypothesis class is given by $\mathcal{H} = \{h_w \mid w \in \mathbb{R}^2, \|w\|_2 = 1\}$, where each hypothesis is defined as $h_w(x) := \text{sgn}(\langle w, x \rangle)$. The data distribution $\mathcal{D}$ is uniform on $\mathcal{X}$, i.e., $\mathcal{D} = U(\mathcal{X})$. Additionally, let $B_w(\alpha) := \{x \in \mathcal{X} \mid |\angle(x, w)| \leq \alpha\}$ denote the set of points within an angular distance up to $\alpha$ to $w$.

**Fully Homomorphic Encryption (FHE) (Appendix F).** FHE (Gentry, 2009) allows for computation on encrypted data *without* decrypting it. An FHE scheme allows to encrypt $x$ via an efficient procedure $e_x = \text{FHE.ENC}(x)$, so that later, for any algorithm $C$, it is possible to run $C$ on $x$ *homomorphically*. More concretely, it is possible to produce an encryption of the result of running $C$ on $x$, i.e., $e_{C,x} := \text{FHE.EVAL}(C, e_x)$. Finally, there is a procedure FHE.DEC that, when given a *secret key* sk, can decrypt $e_{C,x}$, i.e., $y := \text{FHE.DEC}(\text{sk}, e_{C,x})$, where $y$ is the result of running $C$ on $x$. Crucially, encryptions of any two messages are indistinguishable for all efficient adversaries.

**Cryptography-based Learning Task $\mathcal{L}^{\text{crypto}}$ (Figure 5).** $\mathcal{L}^{\text{crypto}}$ is derived from *Lines on Circle Learning Task $\mathcal{L}^\circ$*. Let $w \in \mathcal{X}$. We define the distribution as an equal mixture of two parts $\mathcal{D} = \frac{1}{2}\mathcal{D}_{\text{CLEAR}} + \frac{1}{2}\mathcal{D}_{\text{ENC}}$. The first part, i.e., $\mathcal{D}_{\text{CLEAR}}$, is equal to $x \sim U(\mathcal{X})$ with label $y = h_w(x)$. The second part, i.e., $\mathcal{D}_{\text{ENC}}$, is equal to $x' \sim U(\mathcal{X}), y' = h_w(x'), (x, y) = (\text{FHE.ENC}(x'), \text{FHE.ENC}(y'))$, which can be thought of as $\mathcal{D}_{\text{CLEAR}}$ under an encryption. See Figure 5 for a visual representation.

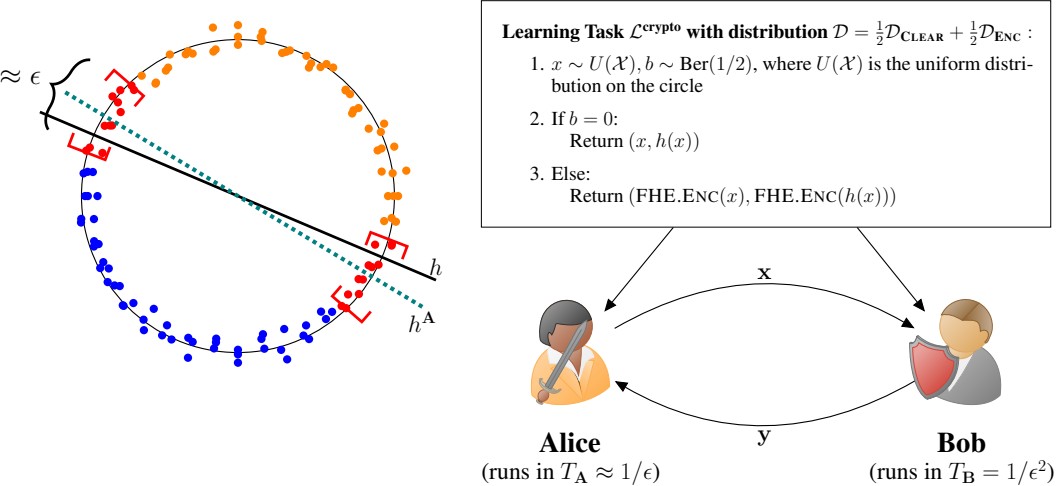

Figure 5: The left part of the figure represents a *Lines on Circle Learning Task $\mathcal{L}^\circ$* with a ground truth function denoted by $h$. On the right, we define a *cryptography-augmented* learning task derived from $\mathcal{L}^\circ$. In its distribution, a "clear" or an "encrypted" sample is observed with equal probability. Given their respective times, both **A** and **B** are able to learn a low-error classifier $h^{\mathbf{A}}$, $h^{\mathbf{B}}$ respectively, by learning only on the *clear samples*. **A** is able to compute a Transferable Attack by computing an encryption of a point close to the decision boundary of her classifier $h^{\mathbf{A}}$.

**Transferable Attack (Figure 5).** Consider the following attack strategy **A**. First, **A** collects $O(1/\epsilon)$ samples from the distribution $\mathcal{D}_{\text{CLEAR}}$ and learns a classifier $h^{\mathbf{A}}_{w'} \in \mathcal{H}$ that is consistent with these samples. Since the VC-dimension of $\mathcal{H}$ is 2, the hypothesis $h^{\mathbf{A}}_{w'}$ has error at most $\epsilon$ with high probability.[3] Next, **A** samples a point $x_{\text{BND}}$ uniformly at random from a region close to the decision

---

[3] **A** can also evaluate $h^{\mathbf{A}}_{w'}$ homomorphically (i.e., run FHE.EVAL) on FHE.ENC($x$) to obtain FHE.ENC($y$) of error $\epsilon$ on $\mathcal{D}_{\text{ENC}}$ also. This means that **A** is able to learn a low-error classifier on $\mathcal{D}$.

boundary of $h_{w'}^{\mathbf{A}}$, i.e., $x_{\text{BND}} \sim U(B_{w'}(\epsilon))$. Finally, with equal probability, $\mathbf{A}$ sets as an attack $\mathbf{x}$ either FHE.ENC($x_{\text{BND}}$) or a uniformly random point $\mathcal{D}_{\text{CLEAR}} = U(\mathcal{X})$. We claim that $\mathbf{x}^4$ satisfies the properties of a Transferable Attack.

Since $h_{w'}^{\mathbf{A}}$ has low error with high probability, $x_{\text{BND}}$ is a uniformly random point from an arc containing the boundary of $h_w$ (see Figure 5). The running time of $\mathbf{B}$ is upper-bounded by $1/\epsilon^2$, meaning it can only learn a classifier with error $\gtrsim 10\epsilon^2$ (see Lemma 3 for details). $\mathbf{B}$'s can only learn (Lemma 3) a classifier of error, $\gtrsim 10\epsilon^2$. Taking these two facts together, we expect $\mathbf{B}$ to misclassify $x'$ with probability $\approx \frac{1}{2} \cdot \frac{10\epsilon^2}{\epsilon} = 5\epsilon > 2\epsilon$, where the factor $\frac{1}{2}$ takes into account that we send an encrypted sample only half of the time. This implies *transferability*.

Note that $\mathbf{x}$ is encrypted with the same probability as in the original distribution because we send FHE.ENC($x_{\text{BND}}$) and a uniformly random $\mathbf{x} \sim \mathcal{D}_{\text{CLEAR}} = U(\mathcal{X})$ with equal probability. Crucially, FHE.ENC($x_{\text{BND}}$) is indistinguishable, for efficient adversaries, from FHE.ENC($x$) for any other $x \in \mathcal{X}$. This follows from the security of the FHE scheme. Consequently, *undetectability* holds. $\quad\square$

**Note 1.** *We want to emphasize that it is crucial (for our construction) that the distribution has both an encrypted ($\mathcal{D}_{\text{ENC}}$) and an unencrypted part ($\mathcal{D}_{\text{CLEAR}}$). If there was no $\mathcal{D}_{\text{CLEAR}}$, then $\mathbf{A}$ would not be able to generate FHE.ENC($x_{\text{BND}}$). The properties of the FHE would allow $\mathbf{A}$ to learn a low-error classifier $h_{w'}^{\mathbf{A}}$, but only under the FHE encryption. Although $\mathbf{A}$ can produce encryptions of points of her choice, she knows $w'$ only under encryption, so she does not know which point to encrypt! If there was no $\mathcal{D}_{\text{ENC}}$, then everything would happen in the clear and so $\mathbf{B}$ would be able to distinguish $x$'s that appear too close to the boundary.*

### 5.2 Tasks with Transferable Attacks imply Cryptography

In this section, we show that a Transferable Attack for any task implies a *cryptographic primitive*.

#### 5.2.1 EFID pairs

In cryptography, an *EFID pair* (Goldreich, 1990) is a pair of distributions $\mathcal{D}_0, \mathcal{D}_1$, that are **E**fficiently samplable, statistically **F**ar, and computationally **I**ndistinguishable. By a seminal result (Goldreich, 1990), we know that the existence of EFID pairs is equivalent to the existence of *Pseudorandom Generators* (PRG). A PRG is an efficient algorithm which stretches short seeds into longer output sequences such that the output distribution on a uniformly chosen seed is computationally indistinguishable from a uniform distribution. Together with what is known about PRGs, this implies that EFID pairs can be used for tasks in cryptography, including encryption and key generation (Goldreich, 1990).

For two time bounds $T, T'$ we call a pair of distributions $(\mathcal{D}_0, \mathcal{D}_1)$ a $(T, T')$ EFID pair if (i) $\mathcal{D}_0, \mathcal{D}_1$ are samplable in time $T$, (ii) $\mathcal{D}_0, \mathcal{D}_1$ are statistically far, (iii) $\mathcal{D}_0, \mathcal{D}_1$ are indistinguishable for algorithms running in time $T'$.

#### 5.2.2 Tasks with Transferable Attacks imply EFID pairs

The second result of this section shows that any task with a Transferable Attack implies the existence of a type of EFID pair. The proof is deferred to Appendix H.

**Theorem 3** (*Tasks with Transferable Attacks imply EFID pairs, informal*). *For every $\epsilon, T, T' \in \mathbb{N}, T \leq T'$, every learning task $\mathcal{L}$ if there exists $\mathbf{A} \in \text{TRANSFATTACK}\left(\mathcal{L}, \epsilon, T, T'\right)$ and there exists a learner running in time $T$ that, with high probability, learns $f$ such that $err(f) \leq \epsilon$, then there exists a $(T, T')$ EFID pair.*

## 6 Tasks with Watermarks and Adversarial Defenses

In this section, we give examples of tasks with Watermarks and Adversarial Defenses. In the first example, we show that hypothesis classes of bounded VC-dimension have Adversarial Defenses against all attackers. The second example is a learning task of bounded VC-dimension that has

---

[4] In this proof sketch, we have $q = 1$, i.e., $\mathbf{A}$ sends only one $x$ to $\mathbf{B}$. This is not true for the formal scheme.

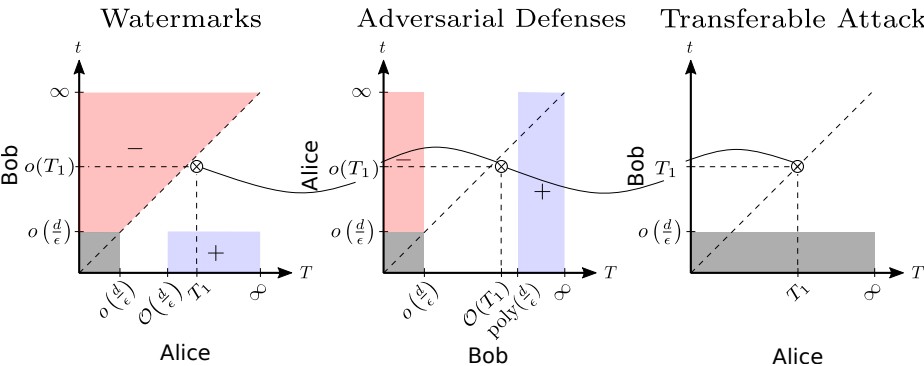

Figure 6: Overview of the taxonomy of learning tasks, illustrating the presence of Watermarks, Adversarial Defenses, and Transferable Attacks for learning tasks of bounded VC dimension. The axes represent the time bound for the parties in the corresponding schemes. The blue regions depict positive results, the red negative, and the gray regimes of parameters which are not of interest. See Lemma 1 and 2 for details about blue regions. The curved line represents a potential application of Theorem 1, which says that at least one of the three points should be blue.

a Watermark, which is secure against fast adversaries. These lemmas demonstrate why the upper bounds on the running time of **A** and **B** are crucial parameters. Lemmas are proven in the appendix.

The first lemma relies heavily on a result from Goldwasser et al. (2020). The authors give a defense against *arbitrary examples* in a transductive model with rejections. In contrast, our model does not allow rejections, but we do require indistinguishability. Careful analysis leads to the following result.

**Lemma 1** (*Adversarial Defense for bounded VC-Dimension, informal*). *Let $d \in \mathbb{N}$ and $\mathcal{H}$ be a binary hypothesis class on input space $\mathcal{X}$ of VC-dimension bounded by $d$. There exists an algorithm* **B** *such that for every $\epsilon \in \left(0, \frac{1}{8}\right)$, $\mathcal{D}$ over $\mathcal{X}$ and $h \in \mathcal{H}$ we have*

$$\mathbf{B} \in \textsc{Defense}\left((\mathcal{D}, h), \epsilon, T_{\mathbf{A}} = \infty, T_{\mathbf{B}} = \texttt{poly}\left(\frac{d}{\epsilon}\right)\right).$$

Note that, by the PAC learning bound, this is a setting of parameters, where **B** has enough time to learn a classifier of error $\epsilon$. By slightly abusing the notation, we write $T_{\mathbf{A}} = \infty$, meaning that the defense is secure against *all* adversaries regardless of their running time.

**Lemma 2** (*Watermark for bounded VC-Dimension against fast Adversaries, informal*). *For every $d \in \mathbb{N}$ there exists a distribution $\mathcal{D}$ and a binary hypothesis class $\mathcal{H}$ of VC-dimension $d$ there exists* **A** *such that for any $\epsilon \in \left(\frac{10000}{d^2}, \frac{1}{8}\right)$ if $h \in \mathcal{H}$ is taken uniformly at random from $\mathcal{H}$ then*

$$\mathbf{A} \in \textsc{Watermark}\left((\mathcal{D}, h), \epsilon, T_{\mathbf{A}} = O\left(\frac{d}{\epsilon}\right), T_{\mathbf{B}} = \frac{d}{100}\right).$$

Note that the setting of parameters is such that **A** can learn (with high probability) a classifier of error $\epsilon$, but **B** is *not* able to learn a low-error classifier in its allotted time $t$. This contrasts with Lemma 5, where **B** has enough time to learn. This is the regime of interest for Watermarks, where the scheme is expected to be secure against fast **B**'s.

## 7 IMPLICATIONS FOR AI SAFETY

In contrast to years of adversarial robustness research (Carlini, 2024), we conjecture that for discriminative learning tasks encountered in safety-critical regimes, an Adversarial Defense *will* exist in the future. Three pieces of evidence support this contrarian belief. (i) Theorem 1, (ii) in the security-critical scenarios for Watermarks, the security should hold even against strong defenders, i.e., $T_{\mathbf{B}}$ approaching $T_{\mathbf{A}}$. In this regime, we believe an analog of Theorem 8 can be shown for Watermarks, given the similarity between the *unremovability* (Definition 1) and *transferability* (Definition 3) property. (iii) Transferable Attacks imply cryptography (Theorem 8), which we suspect is rare in practical scenarios.

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

# A ADDITIONAL METHODS IN RELATED WORK

This section provides an overview of the main areas relevant to our work: Watermarking techniques, adversarial defenses, and transferable attacks on Deep Neural Networks (DNNs). Each subsection outlines important contributions and the current state of research in these areas, offering additional context and details beyond those covered in the main body

## A.1 WATERMARKING

Watermarking techniques are crucial for protecting the intellectual property of machine learning models. These techniques can be broadly categorized based on the type of model they target. We review watermarking schemes for both discriminative and generative models, with a primary focus on discriminative models, as our work builds upon these methods.

### A.1.1 WATERMARKING SCHEMES FOR DISCRIMINATIVE MODELS

Discriminative models, which are designed to categorize input data into predefined classes, have been a major focus of watermarking research. The key approaches in this domain can be divided into black-box and white-box approaches.

**Black-Box Setting.** In the black-box setting, the model owner does not have access to the internal parameters or architecture of the model, but can query the model to observe its outputs. This setting has seen the development of several watermarking techniques, primarily through backdoor-like methods.

Adi et al. (2018) and Zhang et al. (2018) proposed frameworks that embed watermarks using specifically crafted input data (e.g., unique patterns) with predefined outcomes. These watermarks can be verified by feeding these special inputs into the model and checking for the expected outputs, thereby confirming ownership.

Another significant contribution in this domain is by Merrer et al. (2017), who introduced a method that employs adversarial examples to embed the backdoor. Adversarial examples are perturbed inputs that cause the model to produce specific outputs, thus serving as a watermark.

Namba & Sakuma (2019) further enhanced the robustness of black-box watermarking schemes by developing techniques that withstand various model modifications and attacks. These methods ensure that the watermark remains intact and detectable even when the model undergoes transformations.

Provable undetectability of backdoors was achieved in the context of classification tasks by Goldwasser et al. (2022). Unfortunately, it is known ((Goldwasser et al., 2022)) that some undetectable watermarks are easily removed by simple mechanisms similar to randomized smoothing (Cohen et al., 2019).

The popularity of black-box watermarking is due to its practical applicability, as it does not require access to the model's internal workings. This makes it suitable for scenarios where models are deployed as APIs or services. Our framework builds upon these black-box watermarking techniques.

**White-Box Setting.** In contrast, the white-box setting assumes that the model owner has full access to the model's parameters and architecture, allowing for direct examination to confirm ownership. The initial methodologies for embedding watermarks into the weights of DNNs were introduced by Uchida et al. (2017) and Nagai et al. (2018). Uchida et al. (2017) presented a framework for embedding watermarks into the model weights, which can be examined to confirm ownership.

An advancement in white-box watermarking is provided by Darvish Rouhani et al. (2019), who developed a technique to embed an $N$-bit ($N \geq 1$) watermark in DNNs. This technique is both *data-* and *model-dependent*, meaning the watermark is activated only when specific data inputs are fed into

the model. For revealing the watermark, activations from intermediate layers are necessary in the case of white-box access, whereas only the final layer's output is needed for black-box scenarios.

Our work does not focus on white-box watermarking techniques. Instead, we concentrate on exploring the interaction between backdoor-like watermarking techniques, adversarial defenses, and transferable attacks. Overall, watermarking through backdooring has become more popular due to its applicability in the black-box setting.

### A.1.2 WATERMARKING SCHEMES FOR GENERATIVE MODELS

Watermarking techniques for generative models have attracted considerable attention with the advent of Large Language Models (LLMs) and other advanced generative models. This increased interest has led to a surge in research and diverse contributions in this area.

**Backdoor-Based Watermarking for Pre-trained Language Models.** In the domain of Natural Language Processing (NLP), backdoor-based watermarks have been increasingly studied for Pre-trained Language Models (PLMs), as exemplified by works such as (Gu et al., 2022) and (Li et al., 2023). These methods leverage rare or common word triggers to embed watermarks, ensuring that they remain robust across downstream tasks and resilient to removal techniques like fine-tuning or pruning. While these approaches have demonstrated promising results in practical applications, they are primarily empirical, with theoretical aspects of watermarking and robustness requiring further exploration.

**Watermarking the Output of LLMs.** Watermarking the generated text of LLMs is critical for mitigating potential harms. Significant contributions in this domain include (Kirchenbauer et al., 2023), who proposed a watermarking framework that embeds signals into generated text that are invisible to humans but detectable algorithmically. This method promotes the use of a randomized set of "green" tokens during text generation, and detects the watermark without access to the language model API or parameters.

Kuditipudi et al. (2023) introduced robust distortion-free watermarks for language models. Their method ensures that the watermark does not distort the generated text, providing robustness against various text manipulations while maintaining the quality of the output.

Zhao et al. (2023a) presented a provable, robust watermarking technique for AI-generated text. This approach offers strong theoretical guarantees for the robustness of the watermark, making it resilient against attempts to remove or alter it without significantly changing the generated text.

However, Zhang et al. (2023) highlighted vulnerabilities in these watermarking schemes. Their work demonstrates that current watermarking techniques can be effectively broken, raising important considerations for the future development of robust and secure watermarking methods for LLMs.

**Image Generation Models.** Various watermarking techniques have been developed for image generation models to address ethical and legal concerns. Fernandez et al. (2023) introduced a method combining image watermarking with Latent Diffusion Models, embedding invisible watermarks in generated images for future detection. This approach is robust against modifications such as cropping. Wen et al. (2023b) proposed Tree-Ring Watermarking, which embeds a pattern into the initial noise vector during sampling, making the watermark robust to transformations like convolutions and rotations. Jiang et al. (2023) highlighted vulnerabilities in watermarking schemes, showing that human-imperceptible perturbations can evade watermark detection while maintaining visual quality. Zhao et al. (2023c) provided a comprehensive analysis of watermarking techniques for Diffusion Models, offering a recipe for efficiently watermarking models like Stable Diffusion, either through training from scratch or fine-tuning. Additionally, Zhao et al. (2023b) demonstrated that invisible watermarks are vulnerable to regeneration attacks that remove watermarks by adding random noise and reconstructing the image, suggesting a shift towards using semantically similar watermarks for better resilience.

**Audio Generation Models.** Watermarking techniques for audio generators have been developed for robustness against various attacks. Erfani et al. (2017) introduced a spikegram-based method, embedding watermarks in high-amplitude kernels, robust against MP3 compression and other attacks

while preserving quality. Liu et al. (2023) proposed DeAR, a deep-learning-based approach resistant to audio re-recording (AR) distortions.

## A.2 Adversarial Defense

The field of adversarial robustness has a rich and extensive literature (Szegedy et al., 2014; Gilmer et al., 2018; Raghunathan et al., 2018; Wong & Kolter, 2018; Engstrom et al., 2017). Adversarial defenses are essential for ensuring the security and reliability of machine learning models against adversarial attacks that aim to deceive them with carefully crafted inputs.

For discriminative models, there has been significant progress in developing adversarial defenses. Techniques such as adversarial training (Madry et al., 2018), which involves training the model on adversarial examples, have shown promise in improving robustness. Certified defenses (Raghunathan et al., 2018) provide provable guarantees against adversarial attacks, ensuring that the model's predictions remain unchanged within a specified perturbation bound. Additionally, methods like *randomized smoothing* (Cohen et al., 2019) offer robustness guarantees.

A particularly relevant work for our study is (Goldwasser et al., 2020), which considers a different model for generating adversarial examples. This approach has significant implications for the robustness of watermarking techniques in the face of adversarial attacks.

In the context of Large Language Models (LLMs), there is a rapidly growing body of research focused on identifying adversarial examples (Zou et al., 2023; Carlini et al., 2023; Wen et al., 2023a). This research is closely related to the notion of *jailbreaking* (Andriushchenko et al., 2024; Chao et al., 2023; Mehrotra et al., 2024; Wei et al., 2023), which involves manipulating models to bypass their intended constraints and protections.

## A.3 Transferable Attacks and Transductive Learning

Transferable attacks refer to adversarial examples that are effective across multiple models. Moreover, *transductive learning* has been explored as a means to enhance adversarial robustness, and since our Definition 3 captures some notion of transductive learning in the context of Transferable Attacks, we highlight significant contributions in these areas.

**Adversarial Robustness via Transductive Learning.** Transductive learning (Gammerman et al., 1998) has shown promise in improving the robustness of models by utilizing both training and test data during the learning process. This approach aims to make models more resilient to adversarial perturbations encountered at test time.

One significant contribution is by Goldwasser et al. (2020), which explores learning guarantees in the presence of arbitrary adversarial test examples, providing a foundational framework for transductive robustness. Another notable study by Chen et al. (2021) formalizes transductive robustness and proposes a bilevel attack objective to challenge transductive defenses, presenting both theoretical and empirical support for transductive learning's utility.

Additionally, Montasser et al. (2022) introduce a transductive learning model that adapts to perturbation complexity, achieving a robust error rate proportional to the VC dimension. The method by Wu et al. (2020) improves robustness by dynamically adjusting the network during runtime to mask gradients and cleanse non-robust features, validated through experimental results. Lastly, Tramer et al. (2020) critique the standard of adaptive attacks, demonstrating the need for specific tuning to effectively evaluate and enhance adversarial defenses.

**Transferable Attacks on DNNs.** Transferable attacks exploit the vulnerability of models to adversarial examples that generalize across different models. For discriminative models, significant works include Liu et al. (2016), which investigates the transferability of adversarial examples and their effectiveness in black-box attack scenarios, (Xie et al., 2018), who propose input diversity techniques to enhance the transferability of adversarial examples across different models, and (Dong et al., 2019), which presents translation-invariant attacks to evade defenses and improve the effectiveness of transferable adversarial examples.

In the context of generative models, including large language models (LLMs) and other advanced generative architectures, relevant research is rapidly emerging, focusing on the transferability of adversarial attacks. This area is crucial as it aims to understand and mitigate the risks associated with adversarial examples in these powerful models. Notably, Zou et al. (2023) explored universal and transferable adversarial attacks on aligned language models, highlighting the potential vulnerabilities and the need for robust defenses in these systems.

| | | **Undetectability** | **Unremovability** | **Uniqueness** |
|---|---|---|---|---|
| **Classification** | Goldwasser et al. (2022) | ✔ | robust to some smoothing attacks | ✔(E) |
| | Adi et al. (2018); Zhang et al. (2018) | ✔(E) | ✗ | ✔(E) |
| | Merrer et al. (2017) | ✔(E) | robust to fine tunning attacks | ✔(E) |
| **LLMs** | Christ et al. (2023); Kuditipudi et al. (2023) | ✔ | ✗ | ✔ |
| | Zhao et al. (2023a) | ✗ | robust to edit distance attacks only | ✔ |
| | Tiffany Hsu (2023) | ✔(E) | ✗ | ✔ |
| | Kirchenbauer et al. (2023) | ✗ | ✗ | ✔ |

Table 1: Overview of properties across various watermarking schemes. The symbol ✔ denotes properties with formal guarantees or where proof is plausible, whereas ✗ indicates the absence of such guarantees. Entries marked with ✔(E) represent properties observed empirically; these lack formal proof in the corresponding literature, suggesting that deriving such proof may present substantial challenges. The LLM watermarking schemes refer to those applied to text generated by these models.

## B  PRELIMINARIES

**Learning.**   For a set $\Omega$, we write $\Delta(\Omega)$ to denote the set of all probability measures defined on the measurable space $(\Omega, \mathcal{F})$, where $\mathcal{F}$ is some fixed $\sigma$-algebra that is implicitly understood. We denote by $\mathcal{X}$ the domain and by $\mathcal{Y}$ the label space. A *model* is a function $f : \mathcal{X} \to \mathcal{Y}$.

**Definition 4** (*Learning task*). For a fixed $\mathcal{X}, \mathcal{Y}$ a *learning task* is an element of $\Delta\left(\Delta(\mathcal{X}) \times \mathcal{Y}^{\mathcal{X}}\right)$. We will often use $\mathbb{L}$ to denote a learning task.

For a *distribution* $\mathcal{D} \in \Delta(\mathcal{X})$ and a *ground truth* $h : \mathcal{X} \to \mathcal{Y}$, we define an *error* of $f$ as $\mathrm{err}_{\mathcal{D},h}(f) := \mathbb{E}_{x \sim \mathcal{D}}[f(x) \neq h(x)]$, where the index of err will often be understood implicitly and omitted in notation. For $\mathcal{D} \in \Delta(\mathcal{X}), h : \mathcal{X} \to \mathcal{Y}$ we define an *example oracle* $\mathrm{Ex}(\mathcal{D}, h)$ as an oracle that samples $x \sim \mathcal{D}$ and returns $(x, h(x))$.

**Communication.**   When $\mathrm{Ex}(\mathcal{D}, h)$ generates $(x, h(x))$ it is encoded as a bit-string of some length. For a *message space* $\mathcal{M}$ a *representation class* over $(\mathcal{X}, \mathcal{Y})$ is a mapping $R : \mathcal{M} \to \mathcal{Y}^{\mathcal{X}}$.

**Computation.**   Let $U$ be a universal Turing Machine.

### B.1  DISCUSSION

Definition 4 models a learner's prior knowledge of the learning task as a distribution over pairs $(\mathcal{D}, h)$, i.e. over pairs of distributions over the domain $\mathcal{X}$ and ground truths $h : \mathcal{X} \to \mathcal{Y}$. It can be viewed as a generalization of, for instance, PAC-Bayes, where priors are distributions over hypothesis spaces. For us prior knowledge (what we call a learning task) is a distribution over not only hypotheses but also distributions themselves. Note that we consider a realizable scenario as there is a fixed ground truth. We could have considered a more general case, i.e. agnostic learning, where a learning task

would be an element of $\Delta\left(\Delta(\mathcal{X}\times\mathcal{Y})\right)$. We chose the former for simplicity and we believe most of the results would generalize to the agnostic case.

When $\text{Ex}(\mathcal{D},h)$ generates $(x,h(x))$ it is encoded is some form, e.g. $x\in\{0,1\}^n$, but importantly $n$ *is not* a parameter that the learner can control, i.e. the encoding is fixed. This precludes thinking of $n$ as a security parameter that the watermarking party can increase to boost the security.

## C  FORMAL DEFINITIONS

**Definition 5** (*Succinct Circuits*). Let $C$ be a circuit of width $w$ and depth $d$. We will denote $\text{size}(C) := w\cdot d$. We say that $C$ is *succinctly representable* if there exists a circuit of size $100\log(\text{size}(C))^5$ that accepts as input $i\in[w], j, j_1, j_2\in[d], g\in[O(1)]$, where $g$ represents a gate from a universal constant-sized gate set, and returns 0 or 1, depending if $g$ appears in location $(i,j)$ in $C$ and if it is connected to gates in locations $(i-1, j_1)$ and $(i-1, j_2)$.

We are ready to state formal versions of our main definitions.

**Definition 6** (*Watermark*). Let $\mathcal{L} = (\mathcal{D}, h)$ be a learning task. Let $T, t, q\in\mathbb{N}, \epsilon\in\left(0,\frac{1}{2}\right), l, c, s\in (0,1), s < c$, where $t$ bounds the running time of $\mathbf{B}$, and $T$ the running time of $\mathbf{A}$, $q$ the number of queries, $\epsilon$ the risk level, $c$ probability that *uniqueness* holds, $s$ probability that *unremovability* and *undetectability* holds, $l$ the learning probability.

We say that a succinctly representable circuit $\mathbf{A}_{\text{WATERMARK}}$ *of size* $T$ implements a watermarking scheme for $\mathcal{L}$, denoted by $\mathbf{A}_{\text{WATERMARK}}\in\text{WATERMARK}(\mathcal{L},\epsilon,q,T,t,l,c,s)$, if an interactive protocol in which $\mathbf{A}_{\text{WATERMARK}}$ computes $(f,\mathbf{x})$, $f:\mathcal{X}\to\mathcal{Y}, \mathbf{x}\in\mathcal{X}^q$, and $\mathbf{B}$ outputs $\mathbf{y} = \mathbf{B}(f,\mathbf{x}), \mathbf{y}\in\mathcal{Y}^q$ satisfies the following

- **Correctness** ($f$ has low error). With probability at least $l$

$$\text{err}(f)\leq\epsilon.$$

- **Uniqueness** (models trained from scratch give low-error answers). There exists a succinctly representable circuit $\mathbf{B}$ of size $T$ such that with probability at least $c$

$$\text{err}(\mathbf{x},\mathbf{y})\leq 2\epsilon.$$

- **Unremovability** (fast $\mathbf{B}$ gives high-error answers). For every succinctly representable circuit $\mathbf{B}$ *of size at most* $t$ we have that with probability at most $s$

$$\text{err}(\mathbf{x},\mathbf{y})\leq 2\epsilon.$$

- **Undetectability** (fast $\mathbf{B}$ cannot detect that they are tested). Distributions $\mathcal{D}^q$ and $\mathbf{x}\sim\mathbf{A}_{\text{WATERMARK}}$ are $\frac{s}{2}$-indistinguishable for a class of succinctly representable circuits $\mathbf{B}$ *of size at most* $t$.

**Definition 7** (*Adversarial Defense*). Let $\mathcal{L} = (\mathcal{D}, h)$ be a learning task. Let $T, t, q\in\mathbb{N}, \epsilon\in\left(0,\frac{1}{2}\right), l, c, s\in (0,1), s < c$, where $t$ bounds the running time of $\mathbf{A}$, and $T$ the running time of $\mathbf{B}$, $q$ the number of queries, $\epsilon$ the error parameter, $c$ the completeness, $s$ the soundness, $l$ the learning probability.

We say that a succinctly representable circuit $\mathbf{B}_{\text{DEFENSE}}$ of size $T$ implements an adversarial defense for $\mathcal{L}$, denoted by $\mathbf{B}_{\text{DEFENSE}}\in\text{DEFENSE}(\mathcal{L},\epsilon,q,t,T,l,c,s)$, if an interactive protocol in which $\mathbf{B}_{\text{DEFENSE}}$ computes $f:\mathcal{X}\to\mathcal{Y}$, $\mathbf{A}$ replies with $\mathbf{x} = \mathbf{A}(f), \mathbf{x}\in\mathcal{X}^q$, and $\mathbf{B}_{\text{DEFENSE}}$ outputs $b = \mathbf{B}_{\text{DEFENSE}}(f,\mathbf{x}), b\in\{0,1\}$ satisfies the following.

- **Correctness** ($f$ has low error). With probability at least $l$

$$\text{err}(f)\leq\epsilon.$$

---

[5]Constant 100 is chosen arbitrarily. One often considers circuits representable by `polylog`-sized circuits. But for us, the constants play a role and this is why we formulate Definition 5.

- **Completeness** (if $\mathbf{x}$ came from the right distribution $\mathbf{B}_{\text{DEFENSE}}$ does not signal it is attacked). When $\mathbf{x} \sim \mathcal{D}^q$ then with probability at least $c$

$$b = 0.$$

- **Soundness** (fast attacks creating $\mathbf{x}$ on which $f$ makes mistakes are detected). For every succinctly representable circuit $\mathbf{A}$ of size at most $t$ we have that with probability at most $s$,

$$\text{err}(\mathbf{x}, f(\mathbf{x})) > 7\epsilon \text{ and } b = 0.$$

**Definition 8** (*Transferable Attack*). Let $\mathcal{L} = (\mathcal{D}, h)$ be a learning task. Let $T, t, q \in \mathbb{N}, \epsilon \in \left(0, \frac{1}{2}\right), c, s \in (0, 1)$, where $T$ bounds the running time of $\mathbf{A}$ and $\mathbf{B}$, $q$ the number of queries, $\epsilon$ the error parameter, $c$ *transferability* probability, $s$ the *undetectability* probability.

We say that a succinctly representable circuit $\mathbf{A}$ *running in time* $T$ is a transferable adversarial attack, denoted by $\mathbf{A}_{\text{TRANSFATTACK}} \in \text{TRANSFATTACK}(\mathcal{L}, \epsilon, q, T, t, c, s)$, if an interactive protocol in which $\mathbf{A}_{\text{TRANSFATTACK}}$ computes $\mathbf{x} \in \mathcal{X}^q$, and $\mathbf{B}$ outputs $\mathbf{y} = \mathbf{B}(\mathbf{x}), \mathbf{y} \in \mathcal{Y}^q$ satisfies the following.

- **Transferability** (fast provers return high error answers). For every succinctly representable circuit $\mathbf{B}$ of size at most $t$ we have that with probability at least $c$

$$\text{err}(\mathbf{x}, \mathbf{y}) > 2\epsilon.$$

- **Undetectability** (fast provers cannot detect that they are tested). Distributions $\mathbf{x} \sim \mathcal{D}^q$ and $\mathbf{x} := \mathbf{A}_{\text{TRANSFATTACK}}$ are $\frac{s}{2}$-indistinguishable for a class of succinctly representable circuits $\mathbf{B}$ of size at most $t$.

## D  MAIN THEOREM

Before proving our main theorem we recall a result from Lipton & Young (1994a) about simple strategies for large zero-sum games.

**Game theory.**  A *two-player zero-sum game* is specified by a payoff matrix $\mathcal{G}$. $\mathcal{G}$ is an $r \times c$ matrix. MIN, the row player, chooses a probability distribution $p_1$ over the rows. MAX, the column player, chooses a probability distribution $p_2$ over the columns. A row $i$ and a column $j$ are drawn from $p_1$ and $p_2$ and MIN pays $\mathcal{G}_{ij}$ to MAX. MIN tries to minimize the expected payment; MAX tries to maximize it.

By the Min-Max Theorem, there exist optimal strategies for both MIN and MAX. Optimal means that playing first and revealing one's mixed strategy is not a disadvantage. Such a pair of strategies is also known as a Nash equilibrium. The expected payoff when both players play optimally is known as the value of the game and is denoted by $\mathcal{V}(\mathcal{G})$.

We will use the following theorem from Lipton & Young (1994a), which says that optimal strategies can be approximated by uniform distributions over sets of pure strategies of size $O(\log(c))$.

**Theorem 4** (Lipton & Young (1994a)). *Let $\mathcal{G}$ be an $r \times c$ payoff matrix for a two-player zero-sum game. For any $\eta \in (0, 1)$ and $k \geq \frac{\log(c)}{2\eta^2}$ there exists a multiset of pure strategies for the MIN (row player) of size $k$ such that a mixed strategy $p_1$ that samples uniformly from this multiset satisfies*

$$\max_j \sum_i p_1(i)\mathcal{G}_{ij} \leq \mathcal{V}(\mathcal{G}) + \eta(\mathcal{G}_{max} - \mathcal{G}_{min}),$$

*where $\mathcal{G}_{max}, \mathcal{G}_{min}$ denote the maximum and minimum entry of $\mathcal{G}$ respectively. The symmetric result holds for the MAX player.*

**Succinct Representations.**  Before we prove the main theorem we give a short discussion about why we consider succinctly representable circuits. Additionally, we require that the algorithms $\mathbf{A}$ and $\mathbf{B}$ in all the schemes to be *succinctly* representable, meaning their code should be much smaller than their running time. This requirement forbids a trivial way to circumvent learning by *hard-coding* ground-truth classifier in the description of the Watermark or Adversarial Defense algorithms.[6]

---

[6]It is known in certain prover-verifier games to verify classification, described by Anil et al. (2021), this situation leads to undesirable equilibria, which is dubbed as the "trivial verifier" failure mode.

Additionally, the succinct representation of algorithms is also in accordance with how learning takes place in practice, for instance, consider DNNs and learning algorithms for those DNNs. The code representing gradient descent algorithms is almost always much shorter than the time required for the optimization of weights. For instance, a provable neural network model that learns succinct algorithms is described by Goel et al. (2022).

We are ready to prove our main theorem.

**Theorem 5.** *For every learning task* $\mathcal{L} = (\mathcal{D}, h)$*; and* $\epsilon \in (0, \frac{1}{2})$*,* $T, q \in \mathbb{N}$*, such that there exists a succinctly representable circuit of size* $T^{\frac{1}{2^{10}\sqrt{\log(T)}}}$ *that learns* $\mathcal{L}$ *up to error* $\epsilon$ *with probability* $1 - \frac{1}{48}$*, at least one of*

$$\text{WATERMARK}\left(\mathcal{L}, \epsilon, q, T, T^{\frac{1}{2^{10}\sqrt{\log(T)}}}, l = \frac{10}{24}, c = \frac{21}{24}, s = \frac{19}{24}\right),$$

$$\text{DEFENSE}\left(\mathcal{L}, \epsilon, q, T^{\frac{1}{2^{10}\sqrt{\log(T)}}}, 2T, l = 1 - \frac{1}{48}, c = \frac{13}{24}, s = \frac{11}{24}\right),$$

$$\text{TRANSFATTACK}\left(\mathcal{L}, \epsilon, q, T, T, c = \frac{3}{24}, s = \frac{19}{24}\right)$$

*exists.*

*Proof of Theorem 5.* Let $\mathcal{L} = \left(\mathcal{D}, h\right)$ be a learning task. Let $T, q, C \in \mathbb{N}, \epsilon \in \left(0, \frac{1}{2}\right)$.

Let $\mathfrak{Candidate}_{\mathfrak{M}}$ be a set of $T^{\frac{1}{2^{10}\sqrt{\log(T)}}}$-sized succinctly representable circuits computing $(f, \mathbf{x})$, where $f \colon \mathcal{X} \to \mathcal{Y}$. Similarly, let $\mathfrak{Candidate}_{\mathfrak{D}}$ be a set of $T^{\frac{1}{2^{10}\sqrt{\log(T)}}}$-sized succinctly representable circuits accepting as input $(f, \mathbf{x})$ and outputting $(\mathbf{y}, b)$, where $\mathbf{y} \in \mathcal{Y}^q, b \in \{0, 1\}$. We interpret $\mathfrak{Candidate}_{\mathfrak{M}}$ as candidate algorithms for a watermark, and $\mathfrak{Candidate}_{\mathfrak{D}}$ as candidate algorithms for attacks on watermarks.

Define a zero-sum game $\mathcal{G}$ between $(\mathbf{A}, \mathbf{B}) \in \mathfrak{Candidate}_{\mathfrak{M}} \times \mathfrak{Candidate}_{\mathfrak{D}}$. The payoff is given by

$$\mathcal{G}(\mathbf{A}, \mathbf{B}) = \frac{1}{2} \, \mathbb{P}_{(f,\mathbf{x}):=\mathbf{A},(\mathbf{y},b):=\mathbf{B}}\left[\text{err}(f) > \epsilon \text{ or err}(\mathbf{x}, \mathbf{y}) \le 2\epsilon \text{ or } b = 1\right]$$
$$+ \frac{1}{2} \, \mathbb{P}_{f:=\mathbf{A}, \mathbf{x} \sim \mathcal{D}^q, (\mathbf{y},b):=\mathbf{B}}\left[\text{err}(f) > \epsilon \text{ or } \left(\text{err}(\mathbf{x}, \mathbf{y}) \le 2\epsilon \text{ and } b = 0\right)\right],$$

where $\mathbf{A}$ tries to minimize and $\mathbf{B}$ maximize the payoff.

Applying Theorem 4 to $\mathcal{G}$ with $\eta = 2^{-5}$ we get two probability distributions, $p$ over a multiset of pure strategies in $\mathfrak{Candidate}_{\mathfrak{M}}$ and $r$ over a multiset of pure strategies in $\mathfrak{Candidate}_{\mathfrak{D}}$ that lead to a $2^{-5}$-approximate Nash equilibrium.

The size $k$ of the multisets is bounded

$$k \le 2^6 \log\left(|\mathfrak{Candidate}_{\mathfrak{M}}|\right)$$

$$\le 2^6 \log\left(2^{100 \log\left(T^{\frac{1}{2^{10}\sqrt{\log(T)}}}\right)}\right) \qquad \text{Because of the number of possible succinct circuits}$$

$$\le 2^{13} \log\left(T^{\frac{1}{2^{10}\sqrt{\log(T)}}}\right)$$

$$\le 2^3 \sqrt{\log(T)}. \tag{1}$$

Next, observe that the mixed strategy corresponding to the distribution $p$ can be represented by a succinct circuit of size

$$k \cdot 100 \log\left(T^{\frac{1}{2^{10}\sqrt{\log(T)}}}\right) \le \frac{k}{2^3} \sqrt{\log(T)}, \tag{2}$$

because we can create a circuit that is a collection of $k$ circuits corresponding to the multiset of $p$, where each one is of size $100 \log\left(T^{\frac{1}{2^{10}\sqrt{\log(T)}}}\right)$. Combining equation 1 and equation 2 we get that

the size of the circuit succinctly representing the strategy $p$ is bounded by

$$\frac{k}{2^3}\sqrt{\log(T)}$$

$$\leq 2^3\sqrt{\log(T)} \cdot \frac{1}{2^3}\sqrt{\log(T)}$$

$$\leq \log(T).$$

This implies that $p$ can be implemented by a $T$-sized succinctly representable circuit. The same hold for $r$. Let's call the strategy corresponding to $p$, $\mathbf{A}_{\text{Nash}}$, and the strategy corresponding to $r$, $\mathbf{B}_{\text{Nash}}$.

Consider cases:

**Case $\mathcal{G}(\mathbf{A_{NASH}}, \mathbf{B_{NASH}}) \geq \frac{19}{24}$.** Define $\mathbf{B}_{\text{DEFENSE}}$ to work as follows:

1. Simulate $f := \mathbf{L}$, where $\mathbf{L}$ is a succinctly representable circuit of size $T^{\frac{1}{2^{10}\sqrt{\log(T)}}}$, such that

$$\mathbb{P}\left[\text{err}(f) \leq \epsilon\right] \geq 1 - \frac{1}{48}.$$

2. Send $f$ to $\mathbf{A}$.

3. Receive $\mathbf{x}$ from $\mathbf{A}$.

4. Simulate $(\mathbf{y}, b) := \mathbf{B}_{\text{NASH}}(f, \mathbf{x})$.

5. Return $b' = 1$ if $b = 1$ or $d(f(\mathbf{x}), \mathbf{y}) > 3\epsilon q$ and $b' = 0$ otherwise,

where $d(\cdot, \cdot)$ is the Hamming distance. Note that $\mathbf{B}_{\text{DEFENSE}}$ runs in time $2T$ and not $T$ because it first simulates $\mathbf{L}$ and then simulates $\mathbf{B}_{\text{NASH}}$.

We claim that

$$\mathbf{B}_{\text{DEFENSE}} \in \text{DEFENSE}\left(\mathcal{L}, \epsilon, q, T^{\frac{1}{2^{10}\sqrt{\log(T)}}}, 2T, l = 1 - \frac{1}{48}, c = \frac{13}{24}, s = \frac{11}{24}\right). \tag{3}$$

Assume towards contradiction that completeness or soundness of $\mathbf{B}_{\text{DEFENSE}}$ as defined in Definition 7 does not hold.

If completeness of $\mathbf{B}_{\text{DEFENSE}}$ does not hold, then

$$\mathbb{P}_{\mathbf{x} \sim \mathcal{D}^q}\left[b' = 0\right] < \frac{13}{24}. \tag{4}$$

Let us compute the payoff of $\mathbf{A}$, which first runs $f := \mathbf{L}$ and sets $\mathbf{x} \sim \mathcal{D}^q$, in the game $\mathcal{G}$, when playing against $\mathbf{B}_{\text{NASH}}$

$\mathcal{G}(\mathbf{A}, \mathbf{B}_{\text{NASH}})$

$$= \frac{1}{2}\mathbb{P}_{(f,\mathbf{x}):=\mathbf{A}}\left[\text{err}(f) > \epsilon \text{ or } \text{err}(\mathbf{x},\mathbf{y}) \leq 2\epsilon \text{ or } b' = 1\right]$$

$$+ \frac{1}{2}\mathbb{P}_{\substack{f:=\mathbf{A}, \\ \mathbf{x}\sim\mathcal{D}^q}}\left[\text{err}(f) > \epsilon \text{ or } \left(\text{err}(\mathbf{x},\mathbf{y}) \leq 2\epsilon \text{ and } b' = 0\right)\right]$$

$$\leq \delta + \frac{1}{2}\mathbb{P}_{f:=\mathbf{L},\mathbf{x}\sim\mathcal{D}^q}\left[\text{err}(\mathbf{x},\mathbf{y}) \leq 2\epsilon \text{ or } b' = 1\right]$$

$$+ \frac{1}{2}\mathbb{P}_{f:=\mathbf{L},\mathbf{x}\sim\mathcal{D}^q}\left[\text{err}(\mathbf{x},\mathbf{y}) \leq 2\epsilon \text{ and } b' = 0\right] \qquad \text{Def. of } \mathbf{A}, \mathbf{B}_{\text{DEFENSE}}, \mathbb{P}\left[\text{err}(f) \leq \epsilon\right] \geq \frac{47}{48}$$

$$< \frac{1}{48} + \frac{1}{2} + \frac{\frac{13}{24}}{2} \qquad \text{By equation 4}$$

$$= \frac{38}{48}$$

$$\leq \mathcal{G}(\mathbf{A_{NASH}}, \mathbf{B_{NASH}}), \notin$$

where the contradiction is with the properties of Nash equilibria.

Assume that $\mathbf{A}$ breaks the soundness of $\mathbf{B}_{\text{DEFENSE}}$, which translates to

$$\mathbb{P}_{\mathbf{x}:=\mathbf{A}(f)}\left[\text{err}(\mathbf{x}, f(\mathbf{x})) > 7\epsilon \text{ and } b = 0 \text{ and } d(f(\mathbf{x}), \mathbf{y})) > 3\epsilon q\right] > \frac{11}{24}. \tag{5}$$

Let $\mathbf{A}'$ first simulate $f := \mathbf{L}$, then runs $\mathbf{x} := \mathbf{A}(f)$, and returns $(f, \mathbf{x})$. We have

$$\begin{aligned}
&\mathcal{G}(\mathbf{A}', \mathbf{B}_{\text{NASH}}) \\
&= \frac{1}{2} \, \mathbb{P}_{(f,\mathbf{x}):=\mathbf{A}'}\left[\text{err}(f) > \epsilon \text{ or } \text{err}(\mathbf{x}, \mathbf{y}) \le 2\epsilon \text{ or } b' = 1\right] \\
&\quad + \frac{1}{2} \, \mathbb{P}_{f:=\mathbf{A}', \mathbf{x} \sim \mathcal{D}^q}\left[\text{err}(f) > \epsilon \text{ or } \left(\text{err}(\mathbf{x}, \mathbf{y}) \le 2\epsilon \text{ and } b' = 0\right)\right] \\
&= \frac{1}{2} \, \mathbb{P}_{f:=\mathbf{L}, \mathbf{x}=\mathbf{A}(f)}\left[\text{err}(f) > \epsilon \text{ or } \text{err}(\mathbf{x}, \mathbf{y}) \le 2\epsilon \text{ or } b' = 1\right] \\
&\quad + \frac{1}{2} \, \mathbb{P}_{f:=\mathbf{L}, \mathbf{x} \sim \mathcal{D}^q}\left[\text{err}(f) > \epsilon \text{ or } \left(\text{err}(\mathbf{x}, \mathbf{y}) \le 2\epsilon \text{ and } b' = 0\right)\right] \qquad \text{By def. of } \mathbf{A}' \\
&< \frac{1}{2} + \frac{1 - \frac{11}{24}}{2} \qquad \qquad \text{By equation 5} \\
&= \frac{37}{48} \\
&\le \mathcal{G}(\mathbf{A}_{\text{NASH}}, \mathbf{B}_{\text{NASH}}), \, \lightning
\end{aligned}$$

where the contradiction is with the properties of Nash equilibria. Thus equation 3 holds.

**Case $\mathcal{G}(\mathbf{A}_{\text{NASH}}, \mathbf{B}_{\text{NASH}}) < \frac{19}{24}$.** Consider $\mathbf{B}$ that returns $(f(\mathbf{x}), b)$ for a uniformly random $b$. We have

$$\mathcal{G}(\mathbf{A}_{\text{NASH}}, \mathbf{B}) \ge \left(1 - \mathbb{P}_{f:=\mathbf{A}_{\text{NASH}}}\left[\text{err}(f) \le \epsilon\right]\right) + \mathbb{P}_{f:=\mathbf{A}_{\text{Nash}}}\left[\text{err}(f) \le \epsilon\right] \cdot \frac{1}{2},$$

because when $\mathbf{x} \sim \mathcal{D}^q$ and $\text{err}(f) \le \epsilon$ the probability that $\text{err}(\mathbf{x}, \mathbf{y}) \le 2\epsilon$ and $b = 0$ is $\frac{1}{2}$, and similarly when $\mathbf{x} := \mathbf{A}_{\text{NASH}}$ then the probability that $b = 1$ is equal $\frac{1}{2}$. The assumption that $\mathcal{G}(\mathbf{A}_{\text{Nash}}, \mathbf{B}) < \frac{19}{24}$ and properties of Nash equilibria imply that $\mathbb{P}_{f:=\mathbf{A}_{\text{Nash}}}[\text{err}(f) \le \epsilon] \ge \frac{10}{24}$. This implies that *correctness* holds for $\mathbf{A}_{\text{Nash}}$ with $l = \frac{10}{24}$.

Next, assume towards contradiction that *unremovability* of $\mathbf{A}_{\text{NASH}}$ does not hold, i.e., there is $\mathbf{B}$ running in time $t$ such that $\mathbb{P}\left[\text{err}(\mathbf{x}, \mathbf{y}) \le 2\epsilon\right] > \frac{19}{24}$. Consider $\mathbf{B}'$ that on input $(f, \mathbf{x})$ returns $(\mathbf{B}(f, \mathbf{x}), 0)$. Then by definition of $\mathcal{G}$, $\mathcal{G}(\mathbf{A}_{\text{NASH}}, \mathbf{B}') > \frac{19}{24}$, which is a contradiction $\lightning$.

Next, assume towards contradiction that *undetectability* of $\mathbf{A}_{\text{NASH}}$ does not hold, i.e., there exists $\mathbf{B}$ such that it distinguishes $\mathbf{x} \sim \mathcal{D}^q$ from $\mathbf{x} := \mathbf{A}_{\text{NASH}}$ with probability higher than $\frac{19}{24}$. Consider $\mathbf{B}'$ that on input $(f, \mathbf{x})$ returns $(f(\mathbf{x}), \mathbf{B}(f, \mathbf{x}))$.[7] Then by definition of $\mathcal{G}$, $\mathcal{G}(\mathbf{A}_{\text{NASH}}, \mathbf{B}') > \frac{19}{24}$, which is a contradiction $\lightning$

There are two further subcases. If $\mathbf{A}_{\text{NASH}}$ satisfies *uniqueness* then

$$\mathbf{A}_{\text{NASH}} \in \text{WATERMARK}\left(\mathcal{L}, \epsilon, q, T, T^{\frac{1}{2^{10}\sqrt{\log(T)}}}, l = \frac{10}{24}, c = \frac{21}{24}, s = \frac{19}{24}\right).$$

If $\mathbf{A}_{\text{NASH}}$ does not satisfy *uniqueness*, then, by definition, every succinctly representable circuit $\mathbf{B}$ of size $T$ satisfies $\text{err}(\mathbf{x}, \mathbf{y}) \le 2\epsilon$ with probability at most $\frac{21}{24}$. Consider the following $\mathbf{A}$. It computes $(f, \mathbf{x}) := \mathbf{A}_{\text{Nash}}$, ignores $f$ and sends $\mathbf{x}$ to $\mathbf{B}$. By the assumption that *uniqueness* is not satisfied for $\mathbf{A}_{\text{Nash}}$ we have that *transferability* of Definition 3 holds for $\mathbf{A}$ with $c = \frac{3}{24}$. Note that $\mathbf{B}$ in the transferable attack does not receive $f$ but it makes it no easier for it to satisfy the properties. Note that *undetectability* still holds with the same parameter. Thus

$$\mathbf{A}_{\text{NASH}} \in \text{TRANSFATTACK}\left(\mathcal{L}, \epsilon, q, T, T, c = \frac{3}{24}, s = \frac{19}{24}\right).$$

$\square$

---

[7] Formally $\mathbf{B}$ receives as input $(f, \mathbf{x})$ and not only $\mathbf{x}$.

# E  BEYOND CLASSIFICATION

Inspired by Theorem 2, we conjecture a possibility of generalizing our results to generative learning tasks. Instead of a ground truth function, one could consider a ground truth quality oracle $Q$, which measures the quality of every input and output pair. This model introduces new phenomena *not* present in the case of classification. For example, the task of *generation*, i.e., producing a high-quality output $y$ on input $x$, is decoupled from the task of *verification*, i.e., evaluating the quality of $y$ as output for $x$. By decoupled, we mean that there is no clear formal reduction from one task to the other. Conversely, for classification, where the space of possible outputs is small, the two tasks are equivalent. Without going into details, this decoupling is the reason why the proof of Theorem 1 does not automatically transfer to the generative case.

This decoupling introduces new complexities, but it also suggests that considering new definitions may be beneficial. For example, because generation and verification are equivalent for classification tasks, we allowed neither $\mathbf{A}$ nor $\mathbf{B}$ access to $h$, as it would trivialize the definitions. However, a modification of the Definition 6 (Watermark), where access to $Q$ is given to $\mathbf{B}$ could be investigated in the generative case. Interestingly, such a setting was considered in (Zhang et al., 2023), where access to $Q$ was crucial for mounting a provable attack on "all" strong watermarks. As we alluded to earlier, Theorem 2 can be seen as an example of a task, where generation is easy but verification is hard – the opposite to what Zhang et al. (2023) posits.

We hope that careful formalizations of the interaction and capabilities of all parties might give insights into not only the schemes considered in this work, but also problems like weak-to-strong generalization (Burns et al., 2024) or scalable oversight (Brown-Cohen et al., 2023).

# F  FULLY HOMOMORPHIC ENCRYPTION (FHE)

We include a definition of fully homomorphic encryption based on the definition from Goldwasser et al. (2013). The notion of fully homomorphic encryption was first proposed by Rivest, Adleman and Dertouzos Rivest et al. (1978) in 1978. The first fully homomorphic encryption scheme was proposed in a breakthrough work by Gentry in 2009 Gentry (2009). A history and recent developments on fully homomorphic encryption is surveyed in (Vaikuntanathan, 2011).

## F.1  PRELIMINARIES

We say that a function $f$ is *negligible* in an input parameter $\lambda$, if for all $d > 0$, there exists $K$ such that for all $\lambda > K$, $f(\lambda) < \lambda^{-d}$. For brevity, we write: for all sufficiently large $\lambda$, $f(\lambda) = \mathrm{negl}(\lambda)$. We say that a function $f$ is *polynomial* in an input parameter $\lambda$, if there exists a polynomial $p$ such that for all $\lambda$, $f(\lambda) \leq p(\lambda)$. We write $f(\lambda) = \mathrm{poly}(\lambda)$. A similar definition holds for $\mathrm{polylog}(\lambda)$. For two polynomials $p, q$, we say $p \leq q$ if for every $\lambda \in \mathbb{N}$, $p(\lambda) \leq q(\lambda)$.

When saying that a Turing machine $\mathcal{A}$ is p.p.t. we mean that $\mathcal{A}$ is a non-uniform probabilistic polynomial-time machine.

## F.2  DEFINITIONS

**Definition 9** (Goldwasser et al. (2013)). A homomorphic (public-key) encryption scheme FHE is a quadruple of polynomial time algorithms (FHE.KEYGEN, FHE.ENC, FHE.DEC, FHE.EVAL) as follows:

- FHE.KEYGEN$(1^\lambda)$ is a probabilistic algorithm that takes as input the security parameter $1^\lambda$ and outputs a public key $pk$ and a secret key $sk$.

- FHE.ENC$(pk, x \in \{0, 1\})$ is a probabilistic algorithm that takes as input the public key $pk$ and an input bit $x$ and outputs a ciphertext $\psi$.

- FHE.DEC$(sk, \psi)$ is a deterministic algorithm that takes as input the secret key $sk$ and a ciphertext $\psi$ and outputs a message $x^* \in \{0, 1\}$.

- FHE.EVAL$(pk, C, \psi_1, \psi_2, \ldots, \psi_n)$ is a deterministic algorithm that takes as input the public key $pk$, some circuit $C$ that takes $n$ bits as input and outputs one bit, as well as $n$ ciphertexts $\psi_1, \ldots, \psi_n$. It outputs a ciphertext $\psi_C$.

**Compactness:** For all security parameters $\lambda$, there exists a polynomial $p(\cdot)$ such that for all input sizes $n$, for all $x_1, \ldots, x_n$, for all $C$, the output length of FHE.EVAL is at most $p(n)$ bits long.

**Definition 10** (*C-homomorphism, Goldwasser et al. (2013)*). Let $C = \{C_n\}_{n \in \mathbb{N}}$ be a class of boolean circuits, where $C_n$ is a set of boolean circuits taking $n$ bits as input. A scheme FHE is $C$-homomorphic if for every polynomial $n(\cdot)$, for every sufficiently large security parameter $\lambda$, for every circuit $C \in C_n$, and for every input bit sequence $x_1, \ldots, x_n$, where $n = n(\lambda)$,

$$\mathbb{P}\left[\begin{array}{c} (pk, sk) \leftarrow \text{FHE.KEYGEN}(1^\lambda); \\ \psi_i \leftarrow \text{FHE.ENC}(pk, x_i) \text{ for } i = 1 \ldots n; \\ \psi \leftarrow \text{FHE.EVAL}(pk, C, \psi_1, \ldots, \psi_n): \\ \text{FHE.DEC}(sk, \psi) \neq C(x_1, \ldots, x_n) \end{array}\right] = \text{negl}(\lambda),$$

where the probability is over the coin tosses of FHE.KEYGEN and FHE.ENC.

**Definition 11** (*Fully homomorphic encryption*). A scheme FHE is fully homomorphic if it is homomorphic for the class of all arithmetic circuits over $\mathbb{GF}(2)$.

**Definition 12** (*Leveled fully homomorphic encryption*). A leveled fully homomorphic encryption scheme is a homomorphic scheme where FHE.KEYGEN receives an additional input $1^d$ and the resulting scheme is homomorphic for all depth-$d$ arithmetic circuits over $\mathbb{GF}(2)$.

**Definition 13** (*IND-CPA security*). A scheme FHE is IND-CPA secure if for any p.p.t. adversary $\mathcal{A}$,

$$\left| \mathbb{P}\left[(pk, sk) \leftarrow \text{FHE.KEYGEN}(1^\lambda) : \mathcal{A}(pk, \text{FHE.ENC}(pk, 0)) = 1\right] + \right.$$

$$\left. - \mathbb{P}\left[(pk, sk) \leftarrow \text{FHE.KEYGEN}(1^\lambda) : \mathcal{A}(pk, \text{FHE.ENC}(pk, 1)) = 1\right] \right| = \text{negl}(\lambda).$$

We now state the result of Brakerski, Gentry, and Vaikuntanathan (Brakerski et al., 2012) that shows a leveled fully homomorphic encryption scheme based on a standard assumption in cryptography called Learning with Errors (Regev, 2005):

**Theorem 6** (*Fully Homomorphic Encryption, definition from Goldwasser et al. (2013)*). *Assume that there is a constant $0 < \epsilon < 1$ such that for every sufficiently large $\ell$, the approximate shortest vector problem gapSVP in $\ell$ dimensions is hard to approximate to within a $2^{O(\ell^\epsilon)}$ factor in the worst case. Then, for every $n$ and every polynomial $d = d(n)$, there is an IND-CPA secure d-leveled fully homomorphic encryption scheme where encrypting $n$ bits produces ciphertexts of length $poly(n, \lambda, d^{1/\epsilon})$, the size of the circuit for homomorphic evaluation of a function $f$ is $size(C_f) \cdot poly(n, \lambda, d^{1/\epsilon})$ and its depth is $depth(C_f) \cdot poly(\log n, \log d)$.*

## G TRANSFERABLE ATTACKS EXIST

**Learning Theory Preliminaries.** For the next lemma, we will consider a slight generalization of learning tasks to the case where there are many valid outputs for a given input. This can be understood as the case of generative tasks. We call a function $h : \mathcal{X} \times \mathcal{Y} \to \{0, 1\}$ an error oracle for a learning task $(\mathcal{D}, h)$ if the error of $f : \mathcal{X} \to \mathcal{Y}$ is defined as

$$\text{err}(f) := \mathbb{E}_{x \sim \mathcal{D}}[h(x, f(x))],$$

where the randomness of expectation includes the potential randomness of $f$. We assume that all parties have access to samples $(x, y) \in \mathcal{X} \times \mathcal{Y}$, where $x \sim \mathcal{D}$ and $y \in \mathcal{Y}$ is some $y$ such that $h(x, y) = 0$.

The following learning task will be crucial for our construction.

**Definition 14** (*Lines on a Circle Learning Task $\mathcal{L}^\circ$*). The input space is $\mathcal{X} = \{x \in \mathbb{R}^2 \mid \|x\|_2 = 1\}$, and the output space $\mathcal{Y} = \{-1, +1\}$. The hypothesis class is $\mathcal{H} = \{h_w \mid w \in \mathbb{R}^2, \|w\|_2 = 1\}$, where $h_w(x) := \text{sgn}(\langle w, x \rangle)$. Let $\mathcal{D} = U(\mathcal{X})$ and $\mathcal{L} = (\mathcal{D}, \mathcal{H})$. Note that $\mathcal{H}$ has VC-dimension equal to 2 so $\mathcal{L}$ is learnable to error $\epsilon$ with $O(\frac{1}{\epsilon})$ samples.

Moreover, define $B_w(\alpha) := \{x \in \mathcal{X} \mid |\angle(x, w)| \leq \alpha\}$.

**Lemma 3** (*Learning lower bound for $\mathcal{L}^\circ$*). *Let* **L** *be a learning algorithm for $\mathcal{L}^\circ$ (Definition 14) that uses $K$ samples and returns a classifier $f$. Then*

$$\mathbb{P}_{w\sim U(\mathcal{X}), f\leftarrow\mathbf{L}}\left[\mathbb{P}_{x\sim U(\mathcal{X})}[f(x)\neq h_w(x)] \leq \frac{1}{2K}\right] \leq \frac{3}{100}.$$

*Proof.* Consider the following algorithm $\mathcal{A}$. It first simulates **L** on $K$ samples to compute $f$. Next, it performs a smoothing of $f$, i.e., computes

$$f_\eta(x) := \begin{cases} +1, & \text{if } \mathbb{P}_{x'\sim U(B_x(2\pi\eta))}[f(x')=+1] > \mathbb{P}_{x'\sim U(B_x(2\pi\eta))}[f(x')=-1] \\ -1, & \text{otherwise.} \end{cases}$$

Note that if $\text{err}(f) \leq \eta$ for a ground truth $h_w$ then for every $x \in \mathcal{X}\setminus B_x(2\pi\eta)$ we have $f_\eta(x) = h_w(x)$. This implies that $\mathcal{A}$ can be adapted to an algorithm that with probability 1 finds $w'$ such that $|\angle(w, w')| \leq \text{err}(f)$.

Assuming towards contradiction that the statement of the lemma does not hold it means that there is an algorithm using $K$ samples that with probability $\frac{3}{100}$ locates $w$ up to angle $\frac{1}{2K}$.

Consider any algorithm $\mathcal{A}$ using $K$ samples. Probability that $\mathcal{A}$ does not see any sample in $B_w(2\pi\eta)$ is at least

$$(1-4\eta)^K \geq \left((1-4\eta)^{\frac{1}{4\eta}}\right)^{4\eta K} \geq \left(\frac{1}{2e}\right)^{4\eta K},$$

which is bigger than $1 - \frac{3}{100}$ if we set $\eta = \frac{1}{2K}$. But note that if there is no sample in $B_w(2\pi\eta)$ then $\mathcal{A}$ cannot locate $w$ up to $\eta$ with certainty. This proves the lemma. $\square$

**Lemma 4** (*Boosting for $\mathcal{L}^\circ$*). *Let $\eta, \nu \in (0, \frac{1}{4})$, **L** be a learning algorithm for $(\mathcal{D}, \mathcal{H})$ that uses $K$ samples and outputs $f : \mathcal{X} \to \{-1, +1\}$ such that with probability $\delta$*

$$\mathbb{P}_{w\sim U(\mathcal{X}), x\sim U(B_w(2\pi\eta))}[f(x) \neq h_w(x)] \leq \nu. \tag{6}$$

*Then there exists a learning algorithm $\mathbf{L}'$ that uses $\max\left(K, \frac{9}{\eta}\right)$ samples such that with probability $\delta - \frac{1}{1000}$ returns $f'$ such that*

$$\mathbb{P}_{w\sim U(\mathcal{X}), x\sim U(\mathcal{X})}[f'(x) \neq h_w(x)] \leq 4\eta\nu.$$

*Proof.* Let $\mathbf{L}'$ first draws $\max\left(K, \frac{9}{\eta}\right)$ samples $Q$ and defines $g : \mathcal{X} \to \{-1, +1, \perp\}$ as, $g$ maps to $-1$ the smallest continuous interval containing all samples from $Q$ with label $-1$. Similarly $g$ maps to $+1$ the smallest continuous interval containing all samples from $Q$ with label $+1$. The intervals are disjoined by construction. Unmapped points are mapped to $\perp$. Next, $\mathbf{L}'$ simulates **L** with $K$ samples and gets a classifier $f$ that with probability $\delta$ satisfies the assumption of the lemma. Finally, it returns

$$f'(x) := \begin{cases} g(x), & \text{if } g(x) \neq \perp \\ f(x), & \text{otherwise.} \end{cases}$$

Consider 4 arcs defined as the 2 arcs constituting $B_w(2\pi\eta)$ divided into 2 parts each by the line $\{x \in \mathbb{R}^2 \mid \langle w, x \rangle = 0\}$. Let $E$ be the event that some of these intervals do not contain a sample from $Q$. Observe that

$$\mathbb{P}[E] \leq 4(1-\eta)^{\frac{9}{\eta}} \leq \frac{1}{1000}.$$

By the union bound with probability $\delta - \frac{1}{1000}$, $f$ satisfies equation 6 and $E$ does not happen. By definition of $f'$ this gives the statement of the lemma. $\square$

**Theorem 7** (*Transferable Attack for a Cryptography based Learning Task*). *There exists a polynomial $p$ such that for every polynomial $r \geq p^8$ and for every sufficiently large security parameter $\lambda \in \mathbb{N}$ there exists a family of distributions $\mathbb{D}_\lambda = \{\mathcal{D}_\lambda^k\}_k$, hypothesis class of error oracles $\mathcal{H}_\lambda = \{h_\lambda^k\}_k$, distribution $\mathcal{D}_\mathcal{L}$ over $k$ such that the following conditions are satisfied.*

---

[8]This is only a formal requirement so that the interval $(1/r(\lambda), 1/p(\lambda))$ is non-empty.

1. *There exists* $\mathbf{A}$ *such that for all* $\epsilon \in \left( \frac{1}{r(\lambda)}, \frac{1}{p(\lambda)} \right)$ *if* $k \sim \mathcal{D}_{\mathcal{L}}$ *then*

$$\mathbf{A} \in \text{TRANSFATTACK}\left( \left( \mathcal{D}_{\lambda}^{k}, h_{\lambda}^{k} \right), \epsilon, q = \frac{16}{\epsilon}, T = \frac{10^3}{\epsilon^{1.3}}, t = \frac{1}{\epsilon^2}, c = 1 - \frac{1}{10}, s = negl(\lambda) \right).$$

2. *There exists a learner* $\mathbf{L}$ *such that for every* $\epsilon \in \left( \frac{1}{r(\lambda)}, \frac{1}{p(\lambda)} \right)$, *with probability* $1 - \frac{1}{10}$ *over the choice of* $k$ *and the internal randomness of* $\mathbf{L}$, $\mathbf{L}$ *returns a classifier of error at most* $\epsilon$. *Additionally,* $\mathbf{L}$ *runs in time* $\frac{10^3}{\epsilon^{1.3}}$ *and uses* $\frac{900}{\epsilon}$ *samples.*

3. *For every* $\epsilon \in \left( \frac{1}{r(\lambda)}, \frac{1}{p(\lambda)} \right)$, *every learner* $\mathbf{L}$ *using at most* $\frac{1}{\epsilon}$ *samples (and in particular time) the probability over the choice of* $k$ *and the internal randomness of* $\mathbf{L}$ *that it returns a classifier of error at most* $\epsilon$ *is smaller than* $\frac{1}{10}$.

Next, we give a formal proof.

*Proof.* The learning task is based on $\mathcal{L}^{\circ}$ from Definition 14.

**Setting of Parameters for FHE.** Let FHE be a fully homomorphic encryption scheme from Theorem 6. We will use the scheme for constant leveled circuits $d = O(1)$. Let $s(n, \lambda)$ be the polynomial bounding the size of the encryption of inputs of length $n$ with $\lambda$ security as well as bounding size of the circuit for holomorphic evaluation, which is guaranteed to exist by Theorem 6. Let $\beta \in (0, 1)$ and $p$ be a polynomial such that

$$s(n^{\beta}, \lambda, d) \leq (n \cdot p(\lambda))^{0.1}, \tag{7}$$

which exist because $s$ is a polynomial. Let $\lambda \in \mathbb{N}$ and $n := p^{1/\beta}(\lambda)^9$ for the length of inputs in the FHE scheme. Observe

$$
\begin{aligned}
s(n, \lambda, d)) &\leq (p(\lambda) \cdot p(\lambda))^{0.1} && \text{By equation 7} \\
&\leq \frac{1}{\epsilon^{0.2}} && \text{By } \epsilon \in \left( \frac{1}{r(\lambda)}, \frac{1}{p(\lambda)} \right).
\end{aligned} \tag{8}
$$

**Learning Task.** We will omit $\lambda$ from indexes of $\mathcal{D}, \mathbb{D}$ and $h$ for simplicity of notation. Let $\mathbb{D} = \{ \mathcal{D}^{(\text{pk,sk})} \}_{(\text{pk,sk})}, \mathcal{H} = \{ h^{(\text{pk,sk,w})} \}_{(\text{pk,sk,w})}$ indexed by valid public/secret key pairs of FHE and $w \in \mathcal{X}$, with $\mathcal{X}$ as in Definition 14. Let $\mathcal{D}_{\mathcal{L}}$ over $(\text{pk,sk}, w)$ be equal to $\text{FHE.KEYGEN}(1^{\lambda}) \times U(\mathcal{X})$.

For a valid (pk,sk) pair we define $\mathcal{D}^{(\text{pk,sk})}$ as the result of the following process: $x \sim \mathcal{D} = U(\mathcal{X})$, with probability $\frac{1}{2}$ return $(0, x, \text{pk})$ and with probability $\frac{1}{2}$ return $(1, \text{FHE.ENC}(\text{pk}, x), \text{pk})$, where the first element of the triple describes if the $x$ is encrypted or not. $x$ is represented as a number $\in (0, 1)$ using $n$ bits.[10]

For a valid (pk,sk) pair and $w \in \mathcal{X}$ we define $h^{(\text{pk,sk,w})}((b, x, \text{pk}), y)$ as a result of the following process: if $b = 0$ return $\mathbb{1}_{h_w(x)=y}$, otherwise let $x_{\text{DEC}} \leftarrow \text{FHE.DEC}(\text{sk}, x), y_{\text{DEC}} \leftarrow \text{FHE.DEC}(\text{sk}, y)$ and if $x_{\text{DEC}}, y_{\text{DEC}} \neq \bot$ (decryption is succesful) return $\mathbb{1}_{h_w(x_{\text{DEC}})=y_{\text{DEC}}}$ and return 1 otherwise.

**Note 2** ($\Omega(\frac{1}{\epsilon})$-sample learning lower bound.). *Note, that by construction any learner using* $K$ *samples for learning task* $\{ \mathcal{D}^{(\text{pk,sk})} \}_{(\text{pk,sk})}, \{ h^{(\text{pk,sk,w})} \}_{(\text{pk,sk,w})}$ *can be transformed (potentially computationally inefficiently) into a learner using* $K$ *samples for the task from Defnition 14 that returns a classifier of at most the same error. This together with a lower bound for learning from Lemma 3 proves point 3 of the lemma.*

---

[9]Note that this setting allows to represent points on $\mathcal{X}$ up to $2^{-p^{1/\beta}(\lambda)}$ precision and this precision is better than $\frac{1}{r(\lambda)}$ for every polynomial $r$ for sufficiently large $\lambda$. This implies that this precision is enough to allow for learning up to error $\epsilon$, because of the setting $\epsilon \geq \frac{1}{q(\lambda)}$.

[10]Note that the space over which $\mathcal{D}^{(\text{pk,sk})}$ is defined on is *not* $\mathcal{X}$.

---

**Algorithm 1** TRANSFATTACK($\mathcal{D}_\lambda^k, \mathcal{H}_\lambda, \epsilon, \lambda$)

1: **Input:** Oracle access to a distribution $\mathcal{D}_\lambda^k$ for some $\mathcal{D}_\lambda^k \in \mathbb{D}_\lambda$, the hypothesis class $\mathcal{H}_\lambda = \{h_\lambda^k\}_k$, error level $\epsilon \in (0,1)$, and the security parameter $\lambda$.

2: $N := 900/\epsilon, q := 16/\epsilon$
3: $Q = \{((b_i, x_i, \mathrm{pk}), y_i)\}_{i \in [N]} \sim (\mathcal{D}_\lambda^k)^N$ $\quad\quad\quad\quad\quad$ ▷ $N$ i.i.d. samples from $\mathcal{D}_\lambda^k$
4: $Q_{\mathrm{CLEAR}} = \{((b, x, \mathrm{pk}), y) \in Q : b = 0\}$ $\quad\quad\quad$ ▷ $Q_{\mathrm{CLEAR}} \subseteq Q$ of unencrypted $x$'s
5: $f_{w'}(\cdot) := \mathrm{sgn}(\langle w', \cdot \rangle) \leftarrow$ a line consistent with samples from $Q_{\mathrm{CLEAR}}$ ▷ $f_{w'} : \mathcal{X} \to \{-1, +1\}$
6: $\{x_i'\}_{i \in [q]} \sim U(\mathcal{X}^q)$
7: $S \sim U(2^{[q]})$ $\quad\quad\quad\quad\quad\quad\quad\quad\quad\quad\quad$ ▷ $S \subseteq [q]$ a uniformly random subset
8: $E_{\mathrm{BND}} := \emptyset$
9: **for** $i \in [q - |S|]$ **do**
10: $\quad x_{\mathrm{BND}} \sim U(B_{w'}(2\pi(\epsilon + \frac{\epsilon}{100})))$ $\quad\quad$ ▷ $x_{\mathrm{BND}}$ is close to the decision boundary of $f_{w'}$
11: $\quad E_{\mathrm{BND}} := E_{\mathrm{BND}} \cup \{\mathrm{FHE.ENC}(\mathrm{pk}, x_{\mathrm{BND}})\}$
12: **end for**
13: $\mathbf{x} := \{(0, x_i', \mathrm{pk}) \mid i \in [q] \setminus S\} \cup \{(1, x', \mathrm{pk}) \mid x' \in E_{\mathrm{BND}}\}$

14: **Return** $\mathbf{x}$

---

**Definition of A (Algorithm 1).** **A** draws $N$ samples $Q = \{((b_i, x_i, \mathrm{pk}), y_i)\}_{i \in [N]}$ for $N := \frac{900}{\epsilon}$.

Next, **A** chooses a subset $Q_{\mathrm{CLEAR}} \subseteq Q$ of samples for which $b_i = 0$. It trains a classifier $f_{w'}(\cdot) := \mathrm{sgn}(\langle w', \cdot \rangle)$ on $Q_{\mathrm{CLEAR}}$ by returning any $f_{w'}$ consistent with $Q_{\mathrm{CLEAR}}$. This can be done in time

$$N \cdot n \leq \frac{900}{\epsilon} \cdot p^{1/\beta}(\lambda) \leq \frac{900}{\epsilon^{1.1}} \quad\quad\quad (9)$$

by keeping track of the smallest interval containing all samples in $Q_{\mathrm{CLEAR}}$ labeled with $+1$ and then returning any $f_{w'}$ consistent with this interval.

**Note 3** ($O(\frac{1}{\epsilon^{1.3}})$-time learning upper bound.). *First note that* **A** *learns well, i.e., with probability at least* $1 - 2\left(1 - \frac{\epsilon}{100}\right)^{\frac{900}{\epsilon}} \geq 1 - \frac{1}{1000}$ *we have that*

$$|\angle(w, w')| \leq \frac{2\pi\epsilon}{100} \quad\quad\quad (10)$$

*Moreover, $f_{w'}(x)$ can be implemented by a circuit $C_{f_{w'}}$ that compares $x$ with the endpoints of the interval. This can be done by a constant leveled circuit. Moreover $C_{f_{w'}}$ can be evaluated with* FHE.EVAL *in time*

$$size(C_{f_{w'}})s(n, \lambda, d) \leq 10n \cdot s(n, \lambda, d) \leq 10p^{1/\beta}(\lambda)s(n, \lambda, d) \leq \frac{10}{\epsilon^{0.3}},$$

*where the last inequality follows from equation 8. This implies that* **A** *can, in time $T$, return a classifier of error $\leq \epsilon$ for $(\mathcal{D}^{(pk,sk)}, h^{(pk,sk,w)})$. This proves point 2. of the lemma.*

Next, **A** prepares $\mathbf{x}$ as follows. It samples $q = \frac{16}{\epsilon}$ points $\{x_i'\}_{i \in [q]}$ from $\mathcal{X}$ uniformly at random. It chooses a uniformly random subset $S \subseteq [q]$. Next, **A** generates $q - |S|$ inputs using the following process: $x_{\mathrm{BND}} \sim U(B_{w'}(2\pi(\epsilon + \frac{\epsilon}{100})))$ ($x_{\mathrm{BND}}$ is close to the decision boundary of $f_{w'}$), return FHE.ENC($\mathrm{pk}, x_{\mathrm{BND}}$). Call the set of $q - |S|$ points $E_{\mathrm{BND}}$. **A** defines:

$$\mathbf{x} := \{(0, x_i', \mathrm{pk}) \mid i \in [q] \setminus S\} \cup \{(1, x', \mathrm{pk}) \mid x' \in E_{\mathrm{BND}}\}.$$

The running time of this phase is dominated by evaluations of FHE.EVAL, which takes

$$q \cdot s(n, \lambda, d) \leq \frac{16}{\epsilon} \cdot \frac{1}{\epsilon^{0.2}} \leq \frac{16}{\epsilon^{1.2}}, \quad\quad\quad (11)$$

where the first inequality follows from equation 8. Taking the sum of equation 9 and equation 11 we get that the running time of **A** is smaller than the required $T = 10^3/\epsilon^{1.3}$.

**A constitutes a Transferable Attack.**    Now, consider $\mathbf{B}$ that runs in time $t = \frac{1}{\epsilon^2}$. By the assumption $t \leq r(\lambda)$, which implies that the security guarantees of FHE hold for $\mathbf{B}$.

We first claim that $\mathbf{x}$ is indistinguishable from $\mathcal{D}^{(\text{pk,sk})}$ for $\mathbf{B}$. Observe that by construction the distribution of ratio of encrypted and not encrypted $x$'s in $\mathbf{x}$ is identical to that of $\mathcal{D}^{(\text{pk,sk})}$. Moreover, the distribution of unencrypted $x$'s is identical to that of $\mathcal{D}^{(\text{pk,sk})}$ by construction. Finally, by the IND-CPA security of FHE and the fact that the running time of $\mathbf{B}$ is bounded by $q(\lambda)$ for some polynomial $q$ we have that $\text{FHE.ENC}(\text{pk}, x_{\text{BND}})$ is distinguishable from $x \sim \mathcal{X}, \text{FHE.ENC}(\text{pk}, x)$ with advantage at most $\text{negl}(\lambda)$. Thus *undetectability* holds with near perfect soundness $s = \frac{1}{2} + \text{negl}(\lambda)$.

Next, we claim that $\mathbf{B}$ can't return low-error answers on $\mathbf{x}$.

Assume towards contradiction that with probability $\frac{5}{100}$

$$\mathbb{P}_{w \sim U(\mathcal{X}), x \sim U(B_w(2\pi\epsilon))}[f(x) \neq h_w(x)] \leq 10\epsilon. \tag{12}$$

We can apply Lemma 4 to get that there exists a learner using $t + \frac{9}{\epsilon}$ samples that with probability $\frac{4}{100}$ returns $f'$ such that

$$\mathbb{P}_{w \sim U(\mathcal{X}), x \sim U(\mathcal{X})}[f'(x) \neq h_w(x)] \leq 40\epsilon^2. \tag{13}$$

Applying Lemma 3 to equation 13 we know that

$$40\epsilon^2 \geq \frac{1}{2(t + \frac{9}{\epsilon})},$$

which implies

$$t \geq \frac{10}{\epsilon^2},$$

which is a contradiction with the assumed running time of $\mathbf{B}$. Thus equation 12 does not hold and in consequence using equation 10 we have that with probability $1 - \frac{6}{100}$

$$\mathbb{P}_{w \sim U(\mathcal{X}), x \sim U(B_{w'}(2\pi(\epsilon + \frac{\epsilon}{10})))}[f(x) \neq h_w(x)] \geq \frac{10}{14} \cdot 10\epsilon \geq 7\epsilon, \tag{14}$$

where crucially $x$ is sampled from $U(B_{w'})$ and not $U(B_w)$. By Fact 2 we know that $|S| \geq \frac{q}{3}$ with probability at least

$$1 - 2e^{-\frac{q}{72}} = 1 - 2e^{-\frac{1}{8\epsilon}} \geq 1 - \frac{1}{1000}.$$

Another application of the Chernoff bound and the union bound we get from equation 14 that with probability at least $1 - \frac{1}{10}$ we have that $\text{err}(\mathbf{x}, \mathbf{y})$ is larger than $2\epsilon$ by the setting of $q = \frac{16}{\epsilon}$.

$\square$

**Note 4.** *We want to emphasize that it is crucial (for our construction) that the distribution has both an encrypted and an unencrypted part.*

*As mentioned before, if there was no $\mathcal{D}_{\text{CLEAR}}$ then $\mathbf{A}$ would see only samples of the form*

$$(\text{FHE.ENC}(x), \text{FHE.ENC}(y))$$

*and would not know which of them lie close to the boundary of $h_w$, and so it would not be able to choose tricky samples. $\mathbf{A}$ would be able to learn a low-error classifier, but* only *under the encryption. More concretely, $\mathbf{A}$ would be able to homomorphically evaluate a circuit that, given a training set and a test point, learns a good classifier and classifies the test point with it. However, it would* not *be able to, with high probability, generate $\text{FHE.ENC}(x)$, for $x$ close to the boundary as it would not know (in the clear) where the decision boundary is.*

*If there was no $\mathcal{D}_{\text{ENC}}$ then everything would happen in the clear and so $\mathbf{B}$ would be able to distinguish $x$'s that appear too close to the boundary.*

**Fact 2** (*Chernoff-Hoeffding*)**.** *Let $X_1, \ldots, X_k$ be independent Bernoulli variables with parameter $p$. Then for every $0 < \epsilon < 1$*

$$\mathbb{P}\left[\left|\frac{1}{k}\sum_{i=1}^{k} X_i - p\right| > \epsilon\right] \leq 2e^{-\frac{\epsilon^2 k}{2}}$$

and

$$\mathbb{P}\left[\frac{1}{k}\sum_{i=1}^{k} X_i \le (1-\epsilon)p\right] \le e^{-\frac{\epsilon^2 kp}{2}}.$$

Also for every $\delta > 0$

$$\mathbb{P}\left[\frac{1}{k}\sum_{i=1}^{k} X_i > (1+\delta)p\right] \le e^{-\frac{\delta^2 kp}{2+\delta}}$$

## H  TRANSFERABLE ATTACKS IMPLY CRYPTOGRAPHY

### H.1  EFID PAIRS

The typical way in which security of EFID pairs is defined, e.g., in (Goldreich, 1990), is that they should be secure against all polynomial-time algorithms. However, for the case of pseudorandom generators (PRGs), which are known are equivalent to EFIDs pairs, more granular notions of security were considered. For instance, in (Nisan, 1990) the existence of PRGs secure against time and space bounded adversaries was considered. In a similar spirit we consider EFID pairs that are secure against adversaries with a fixed time bound.

**Definition 15** (*Total Variation*). For two distrbutions $\mathcal{D}_0, \mathcal{D}_1$ over a finite domain $\mathcal{X}$ we define their *total variation distance* as

$$\triangle(\mathcal{D}_0, \mathcal{D}_1) := \sum_{x \in \mathcal{X}} \frac{1}{2} |\mathcal{D}_0(x) - \mathcal{D}_1(x)|.$$

**Definition 16** (*EFID pairs*). For parameters $\eta, \delta \in (0,1)$ we call a pair of distributions $(\mathcal{D}_0, \mathcal{D}_1)$ a $(T, T', \eta, \delta)$ EFID pair if

1. $\mathcal{D}_0, \mathcal{D}_1$ are samplable in time $T$,

2. $\triangle(\mathcal{D}_0, \mathcal{D}_1) \ge \eta$,

3. $\mathcal{D}_0, \mathcal{D}_1$ are $\delta$-indistinguishable for adversaries running in time $T'$.

### H.2  TRANSFERABLE ATTACKS IMPLY EFID PAIRS

**Theorem 8** (*Tasks with Transferable Attacks imply EFID pairs*). *For every $\epsilon, T, T' \in \mathbb{N}, T \le T'$, every learning task $\mathcal{L}$ if there exists $\mathbf{A} \in \text{TRANSFATTACK}\left(\mathcal{L}, \epsilon, q, T, T', c, s\right)$ and there exists a learner running in time $T$ that, with probability $p$, learns $f$ such that $err(f) \le \epsilon$, then there exists a $(T, T', \frac{1}{2}(p + c - 1 - e^{-\frac{\epsilon q}{3}}), \frac{s}{2})$ EFID pair.*

*Proof.* Let $\epsilon, T, T', q, c, s, \mathcal{L} = (\mathcal{D}, h)$ and $\mathbf{A}$ be as in the assumption of the theorem. Firstly, define $\mathcal{D}_0 := \mathcal{D}^q$, where $q$ is the number of samples $\mathbf{A}$ sends in the attack. Secondly, define $\mathcal{D}_1$ to be the distribution of $\mathbf{x} := \mathbf{A}$. Note that $\mathbf{x} \in \mathcal{X}^q$.

Observe that $\mathcal{D}_0, \mathcal{D}_1$ are samplable in time $T$ as $\mathbf{A}$ runs in time $T$. Secondly, $\mathcal{D}_0, \mathcal{D}_1$ are $\frac{s}{2}$-indistinguishable for $T'$-bounded adversaries by *undetectability* of $\mathbf{A}$. Finally, the fact that $\mathcal{D}_0, \mathcal{D}_1$ are statistically far follows from *transferability*. Indeed, the following procedure accepting input $\mathbf{x} \in \mathcal{X}^q$ is a distinguisher:

1. Run the learner (the existence of which is guaranteed by the assumption of the theorem) to obtain $f$.

2. $\mathbf{y} := f(\mathbf{x})$.

3. If $err(\mathbf{x}, \mathbf{y}) \le 2\epsilon$ return 0, otherwise return 1.

If $\mathbf{x} \sim \mathcal{D}_0 = \mathcal{D}^q$ then $err(f) \le \epsilon$ with probability $p$. By Fact 2 and the union bound we also know that $err(\mathbf{x}, \mathbf{y}) \le 2\epsilon$ with probability $p - e^{-\frac{\epsilon q}{3}}$ and so, the distinguisher will return 0 with

probability $p - e^{-\frac{\epsilon q}{3}}$. On the other hand, if $\mathbf{x} \sim \mathcal{D}_1 = \mathbf{A}$ we know from *transferability* of $\mathbf{A}$ that every algorithm running in time $T'$ will return $\mathbf{y}$ such that $\text{err}(\mathbf{x}, \mathbf{y}) > 2\epsilon$ with probability at least $c$. By the assumption that $T' \geq T$ we know that $\text{err}(\mathbf{x}, f(\mathbf{x})) > 2\epsilon$ with probability at least $c$ also. Consequently, the distinguisher will return 1 with probability at least $c$ in this case. By the properties of total variation this implies that $\triangle(\mathcal{D}_0, \mathcal{D}_1) \geq \frac{1}{2}(p + c - 1 - e^{-\frac{\epsilon q}{3}})$ Summarizing, $(\mathcal{D}_0, \mathcal{D}_1)$ is a $(T, T', \frac{1}{2}(p + c - 1 - e^{-\frac{\epsilon q}{3}}), \frac{s}{2})$ EFID pair.

**Note 5** (*Setting of parameters*). *Observe that if $p \approx 1$, i.e., it is possible to almost surely learn $f$ in time $T$ such that $\text{err}(f) \leq \epsilon$, $c$ is a constant, $q = \Omega(\frac{1}{\epsilon})$ then $\triangle(\mathcal{D}_0, \mathcal{D}_1)$ is a constant.*

**Note 6.** *We want to emphasize that our distinguisher crucially uses the error oracle in its last step. So it is possible that it is not implementable for all time bounds!*

$\square$

# I    ADVERSARIAL DEFENSES EXIST

Our result is based on (Goldwasser et al., 2020). Before we state and prove our result we give an overview of the learning model considered in (Goldwasser et al., 2020).

## I.1    TRANSDUCTIVE LEARNING WITH REJECTIONS.

In (Goldwasser et al., 2020) the authors consider a model, where a learner $\mathbf{L}$ receives a training set of labeled samples from the original distribution $(\mathbf{x}_\mathcal{D}, \mathbf{y}_\mathcal{D} = h(\mathbf{x}_\mathcal{D})), \mathbf{x} \sim \mathcal{D}^N, \mathbf{y}_\mathcal{D} \in \{-1, +1\}^N$, where $h$ is the ground truth, together with a test set $\mathbf{x}_T \in \mathcal{X}^q$. Next, $\mathbf{L}$ uses $(\mathbf{x}_\mathcal{D}, \mathbf{y}_\mathcal{D}, \mathbf{x}_T)$ to compute $\mathbf{y}_T \in \{-1, +1, \square\}^q$, where $\square$ represents that $\mathbf{L}$ abstains (rejects) from classifying the corresponding $x$.

Before we define when learning is successful, we will need some notation. For $q \in \mathbb{N}, \mathbf{x} \in \mathcal{X}^q, \mathbf{y} \in \{-1, +1, \square\}^q$ we define

$$\text{err}(\mathbf{x}, \mathbf{y}) := \frac{1}{q} \sum_{i \in [q]} \mathbb{1}_{\left\{h(x_i) \neq y_i, y_i \neq \square, h(x_i) \neq \perp\right\}}, \quad \square(\mathbf{y}) := \frac{1}{q}\left|\left\{i \in [q] : y_i = \square\right\}\right|,$$

which means that we count $(x, y) \in \mathcal{X} \times \{-1, +1, \square\}$ as an error if $h$ is well defined on $x$, $y$ is not an abstention and $h(x) \neq y$.

Learning is successful if it satisfies two properties.

- If $\mathbf{x}_T \sim \mathcal{D}^q$ then with high probability $\text{err}(\mathbf{x}_T, \mathbf{y}_T)$ and $\square(\mathbf{y}_T)$ are small.
- For *every* $\mathbf{x}_T \in \mathcal{X}^q$ with high probability $\text{err}(\mathbf{x}_T, \mathbf{y}_T)$ is small.[11]

The formal guarantee of a result from Goldwasser et al. (2020) are given in Theorem 9. Let's call this model Transductive Learning with Rejections (TLR).

Note the differences between TLR and our definition of Adversarial Defenses. To compare the two models we associate the learner $\mathbf{L}$ from TLR with $\mathbf{B}$ in our setup, and the party producing $\mathbf{x}_T$ with $\mathbf{A}$ in our definition. First, in TLR, $\mathbf{B}$ does not send $f$ to $\mathbf{A}$. Secondly, and most importantly, we do not allow $\mathbf{B}$ to reply with rejections ($\square$) but instead require that $\mathbf{B}$ can "distinguish" that it is being tested (see soundness of Definition 7). Finally, there are no apriori time bounds on either $\mathbf{A}$ or $\mathbf{B}$ in TLR. The models are similar but a priori incomparable and any result for TLR needs to be carefully analyzed before being used to prove that it is an Adversarial Defense.

## I.2    FORMAL GUARANTEE FOR TRANSDUCTIVE LEARNING WITH REJECTIONS (TLR)

Theorem 5.3 from Goldwasser et al. (2020) adapted to our notation reads.

---

[11]Note that, crucially, in this case $\square(\mathbf{y}_T)$ might be very high, e.g., equal to 1.

**Theorem 9** (*TLR guarantee (Goldwasser et al. (2020))*)**.** *For any $N \in \mathbb{N}, \epsilon \in (0,1), h \in \mathcal{H}$ and distribution $\mathcal{D}$ over $\mathcal{X}$:*

$$\mathbb{P}_{\mathbf{x}_{\mathcal{D}}, \mathbf{x}'_{\mathcal{D}} \sim \mathcal{D}^N} \left[ \forall \, \mathbf{x}_T \in \mathcal{X}^N : err(\mathbf{x}_T, f(\mathbf{x}_T)) \leq \epsilon^* \wedge \square \left( f\left(\mathbf{x}'_{\mathcal{D}}\right) \right) \leq \epsilon^* \right] \geq 1 - \epsilon,$$

*where $\epsilon^* = \sqrt{\frac{2d}{N} \log\left(2N\right) + \frac{1}{N}\log\left(\frac{1}{\epsilon}\right)}$ and $f = \text{REJECTRON}(\mathbf{x}_{\mathcal{D}}, h(\mathbf{x}_{\mathcal{D}}), \mathbf{x}_T, \epsilon^*)$, where $f : \mathcal{X} \to \{-1, +1, \square\}$ and $d$ denotes the VC-dimension on $\mathcal{H}$. REJECTRON is defined in Figure 2. in (Goldwasser et al., 2020).*

REJECTRON is an algorithm that accepts a labeled training set $(\mathbf{x}_{\mathcal{D}}, h(\mathbf{x}_{\mathcal{D}}))$ and a test set $\mathbf{x}_T$ and returns a classifier $f$, which might reject some inputs. The learning is successful if with a high probability $f$ rejects a small fraction of $\mathcal{D}^N$ and for every $\mathbf{x}_T \in \mathcal{X}^N$ the error on labeled $x$'s in $\mathbf{x}_T$ is small.

### I.3 ADVERSARIAL DEFENSE FOR BOUNDED VC-DIMENSION

We are ready to state the main result of this section.

**Lemma 5** (*Adversarial Defense for bounded VC-dimension*)**.** *Let $d \in \mathbb{N}$ and $\mathcal{H}$ be a binary hypothesis class on input space $\mathcal{X}$ of VC-dimension bounded by $d$. There exists an algorithm $\mathbf{B}$ such that for every $\epsilon \in \left(0, \frac{1}{8}\right)$, $\mathcal{D}$ over $\mathcal{X}$ and $h \in \mathcal{H}$ we have*

$$\mathbf{B} \in \text{DEFENSE}\left((\mathcal{D}, h), \epsilon, q = \frac{d\log^2(d)}{\epsilon^3}, t = \infty, T = \texttt{poly}\left(\frac{d}{\epsilon}\right), l = 1 - \epsilon, c = 1 - \epsilon, s = \epsilon\right).$$

*Proof.* The proof is based on an algorithm from Goldwasser et al. (2020).

**Construction of B.** Let $\epsilon \in (0, 1)$ and

$$N := \frac{d\log^2(d)}{\epsilon^3}.$$

Let $q := N$. First, $\mathbf{B}$, draws $N$ labeled samples $(\mathbf{x}_{\text{FRESH}}, h(\mathbf{x}_{\text{FRESH}}))$. Next, it finds $f \in \mathcal{H}$ consistent with them and sends $f$ to $\mathbf{A}$. Importantly this computation is the same as the first step of REJECTRON.

Next, $\mathbf{B}$ receives as input $\mathbf{x} \in \mathcal{X}^q$ from $\mathbf{A}$. $\mathbf{B}$. Let $\epsilon^* := \sqrt{\frac{2d}{N} \log\left(2N\right) + \frac{1}{N}\log\left(\frac{1}{\epsilon}\right)}$. Next $\mathbf{B}$ runs $f' = \text{REJECTRON}(\mathbf{x}_{\text{FRESH}}, h(\mathbf{x}_{\text{FRESH}}), \mathbf{x}, \epsilon^*)$, where REJECTRON is starting from the second step of the algorithm (Figure 2 (Goldwasser et al., 2020)). Importantly, for every $x \in \mathcal{X}$, if $f'(x) \neq \square$ then $f(x) = f'(x)$. In words, $f'$ is equal to $f$ everywhere where $f'$ does not reject.

Finally $\mathbf{B}$ returns 1 if $\square(f'(\mathbf{x})) > \frac{2}{3}\epsilon$, and returns 0 otherwise.

**B is a Defense.** First, by the standard PAC theorem we have that with probability at least $1 - \epsilon$, $\text{err}(f) \leq \frac{\epsilon}{2}$. This means that *correctness* holds with probability $l = 1 - \epsilon$.

Note that with our setting of $N$, we have that

$$\epsilon^* \leq \frac{\epsilon}{2}.$$

Theorem 9 guarantees that

- if $\mathbf{x} \in \mathcal{D}^q$ then with probability at least $1 - \epsilon$ we have that

$$\square(f'(\mathbf{x})) \leq \frac{\epsilon}{2}.$$

  which in turn implies that with the same probability $\mathbf{B}$ returns $b = 0$. This implies that *completeness* holds with probability $1 - \epsilon$.

- for every $\mathbf{x} \in \mathcal{X}^q$ with probability at least $1 - \epsilon$ we have that

$$\text{err}(\mathbf{x}, f'(\mathbf{x})) \leq \frac{\epsilon}{2}.$$

To compute soundness we want to upper bound the probability that $\text{err}(\mathbf{x}, f(\mathbf{x})) > 2\epsilon^{12}$ and $b = 0$. By construction of $\mathbf{B}$ if $b = 0$ then $\square(f'(\mathbf{x})) \leq \frac{2\epsilon}{3}$, which means that with probability at least $1 - \epsilon$

$$\text{err}(\mathbf{x}, \mathbf{y}) \leq \frac{2\epsilon}{3} + \frac{\epsilon}{2} < 2\epsilon \text{ or } b = 1.$$

This translates to *soundness* holding with $s = \epsilon$.

REJECTRON runs in polynomial time in the number of samples and makes $O(\frac{1}{\epsilon})$ calls to an Empirical Risk Minimizer on $\mathcal{H}$ (that we assume runs in time polynomial in $d$), which implies the promised running time. $\square$

## J WATERMARKS EXIST

**Lemma 6** (*Watermark for bounded VC-dimension against fast adversaries*)**.** *For every $d \in \mathbb{N}$ there exists a distribution $\mathcal{D}$ and a binary hypothesis class $\mathcal{H}$ of VC-dimension $d$ there exists $\mathbf{A}$ such that for any $\epsilon \in \left(\frac{10000}{d}, \frac{1}{8}\right)$ if $h \in \mathcal{H}$ is taken uniformly at random from $\mathcal{H}$ then*

$$\mathbf{A} \in \text{WATERMARK}\left((\mathcal{D}, h), \epsilon, q = O\left(\frac{1}{\epsilon}\right), T = O\left(\frac{d}{\epsilon}\right), t = \frac{d}{100}, l = 1 - \frac{1}{100}, c = 1 - \frac{2}{100}, s = \frac{56}{100}\right).$$

*Proof.* Let $\mathcal{X} = \mathbb{N}$. Let $\mathcal{D}$ be the uniform distribution over $[N]$ for $N = 100d^2$. Let $\mathcal{H}$ be the concept class of functions that have exactly $d$ +1's in $[N]$. Note $\mathcal{H}$ has VC-dimension $d$. Let $h \in \mathcal{H}$ be the ground truth.

**Construction of A.** $\mathbf{A}$ works as follows. It draws $n = O\left(\frac{d}{\epsilon}\right)$ samples from $\mathcal{D}$ labeled with $h$. Let's call them $\mathbf{x}_{\text{TRAIN}}$. Let

$$A := \{x \in [N] : \mathbf{x}_{\text{TRAIN}}, h(x) = +1\}, B := \{x \in [N] : x \in \mathbf{x}_{\text{TRAIN}}, h(x) = -1\}.$$

$\mathbf{A}$ takes a uniformly random subset $A_w \subseteq A$ of size $q$. It defines sets

$$A' := A \setminus A_w, \ B' := B \cup A_w.$$

$\mathbf{A}$ computes $f$ consistent with the training set $\{(x, +1) : x \in A'\} \cup \{(x, -1) : x \in B'\}$. $\mathbf{A}$ samples $S \sim \mathcal{D}^q$. It defines the watermark to be $\mathbf{x} := A_w$ with probability $\frac{1}{2}$ and $\mathbf{x} := S$ with probability $\frac{1}{2}$.

$\mathbf{A}$ sends $(f, \mathbf{x})$ to $\mathbf{B}$. $\mathbf{A}$ can be implemented in time $O\left(\frac{d}{\epsilon}\right)$.

**A is a Watermark.** We claim that $(f, \mathbf{x})$ constitutes a watermark.

It is possible to construct a watermark of prescribed size, i.e., find a subset $A_w$ of a given size, only if $|A| \geq q$. The probability that a single sample from $\mathcal{D}$ is labeled $+1$ is $\frac{d}{N}$, so by the Chernoff bound (Fact 2) $|A|, |B| > \frac{dn}{2N} \geq q$ with probability $1 - \frac{1}{100}$, where we used that $n = O\left(\frac{d}{\epsilon}\right), N = 100d^2, q = O(\frac{1}{\epsilon})$.

**Correctness.** Let $h'(x) := h(x)$ if $x \in [N] \setminus A_w$ and $h'(x) := -h(x)$ otherwise. Note that $h'$ has exactly $d - q$ +1's in $[N]$. By construction, $f$ is a classifier consistent with $h'$. By the PAC theorem we know that with probability $1 - \frac{1}{100}$, $f$ has an error at most $\epsilon$ wrt to $h'$ (because the hypothesis class of functions with *at most* $d$ +1's has a VC dimension of $O(d)$). $h'$ differs from $h$ on $q$ points, so

$$\text{err}(f) \leq \epsilon + q/N = O\left(\epsilon + \frac{1}{\epsilon d^2}\right) = O(\epsilon). \tag{15}$$

with probability $1 - \frac{1}{100}$, which implies that *correctness* is satisfied with $l = 1 - \frac{1}{100}$.

---

[12]Note that we measure the error of $f$ not $f'$.

**Distinguishing of x and $\mathcal{D}^q$.** Note that the distribution of $A_w$ is the same as the distribution of a uniformly random subset of $[N]$ of size $q$ (when taking into account the randomness of the choice of $h \sim U(\mathcal{H})$). Observe that the probability that drawing $q$ i.i.d. samples from $U([N])$ we encounter repetitions is at most

$$\frac{1}{N} + \frac{2}{N} + \cdots + \frac{q}{N} \leq \frac{3q^2}{N} \leq \frac{1}{100},$$

because $q < \frac{d}{100} < \frac{\sqrt{N}}{10}$. This means that $\frac{1}{100}$ is an information-theoretic upper bound on the distinguishing advantage between $\mathbf{x} = A_w$ and $\mathcal{D}^q$.

Moreover, $\mathbf{B}$ has access to at most $t$ samples and the probability that the set of samples $\mathbf{B}$ draws from $\mathcal{D}^t$ and $A_w$ have empty intersection is at least $1 - \frac{1}{100}$. It is because it is at least $(1 - \frac{t}{N})^t \geq (1 - \frac{1}{\sqrt{N}})^{\sqrt{N/10}} \geq 1 - \frac{1}{100}$, where we used that $t < \frac{\sqrt{N}}{10}$.[13]

Note that by construction $f$ maps all elements of $A_w$ to $-1$. The probability over the choice of $F \sim \mathcal{D}^q$ that $F \subseteq h^{-1}(\{-1\})$, i.e., all elements of $F$ have true label $-1$, is at least

$$\left(1 - \frac{d}{N}\right)^q \geq 1 - \frac{1}{100}.$$

The three above observations and the union bound imply that the distinguishing advantage for distinguishing $\mathbf{x}$ from $\mathcal{D}^q$ of $\mathbf{B}$ is at most $\frac{4}{100}$ and so the *undetectability* holds with $s = \frac{8}{100}$.

**Unremovability.** Assume, towards contradiction with *unremovability*, that $\mathbf{B}$ can find $\mathbf{y}$ that with probability $s' = \frac{1}{2} + \frac{6}{100}$ satisfies $\text{err}(\mathbf{x}, \mathbf{y}) \leq 2\epsilon$. Notice, that $\text{err}(A_w, f(A_w)) = 1$ by construction.

Consider an algorithm $\mathcal{A}$ for distinguishing $A_w$ from $\mathcal{D}^q$. Upon receiving $(f, \mathbf{x})$ it first runs $\mathbf{y} = \mathbf{B}(f, \mathbf{x})$ and returns 1 iff $d(\mathbf{y}, f(\mathbf{x})) \geq \frac{q}{2}$. We know that the distinguishing advantage is at most $\frac{1}{2} + \frac{4}{100}$, so

$$\frac{1}{2}\mathbb{P}_{\mathbf{x}:=A_w}[\mathcal{A}(f, \mathbf{x}) = 1] + \frac{1}{2}\mathbb{P}_{\mathbf{x} \sim \mathcal{D}^q}[\mathcal{A}(f, \mathbf{x}) = 0] \leq \frac{1}{2} + \frac{4}{100}.$$

But also note that

$$
\begin{aligned}
s' &\leq \mathbb{P}_{\mathbf{x} \sim \mathbf{A}}[\text{err}(\mathbf{x}, \mathbf{y}) \leq 2\epsilon] \\
&\leq \frac{1}{2}\mathbb{P}_{\mathbf{x}:=A_w}[d(\mathbf{y}, f(\mathbf{x})) \geq (1 - 2\epsilon)q] + \frac{1}{2}\mathbb{P}_{\mathbf{x} \sim \mathcal{D}^q}[d(\mathbf{y}, f(\mathbf{x})) \leq (2\epsilon + \text{err}(f))q] \\
&\leq \frac{1}{2}\mathbb{P}_{\mathbf{x}:=A_w}[d(\mathbf{y}, f(\mathbf{x})) \geq q/2] + \frac{1}{2}\mathbb{P}_{\mathbf{x} \sim \mathcal{D}^q}[d(\mathbf{y}, f(\mathbf{x})) \leq q/2] + \frac{1}{100} \\
&\leq \frac{1}{2}\mathbb{P}_{\mathbf{x}:=A_w}[\mathcal{A}(f, \mathbf{x}) = 1] + \frac{1}{2}\mathbb{P}_{\mathbf{x} \sim \mathcal{D}^q}[\mathcal{A}(f, \mathbf{x}) = 0] + \frac{1}{100}.
\end{aligned}
$$

Combining the two above equations we get a contradiction and thus the *unremovability* holds with $s' = \frac{1}{2} + \frac{6}{100}$.

**Uniqueness.** The following $\mathbf{B}$ certifies *uniqueness*. It draws $O\left(\frac{d}{\epsilon}\right)$ samples from $\mathcal{D}$, let's call them $\mathbf{x}'_{\text{TRAIN}}$ and trains $f'$ consistent with it. By the PAC theorem $\text{err}(f') \leq \epsilon$ with probability at least $1 - \frac{1}{100}$. Next upon receiving $\mathbf{x} \in \mathcal{X}^q = [N]^q$ it returns $y = f'(\mathbf{x})$. By the fact that $\mathbf{x}$ is a random subset of $[N]$ of size $q$ by the Chernoff bound, the union bound we know that $\text{err}(\mathbf{x}, \mathbf{y}) = \text{err}(\mathbf{x}, f'(\mathbf{x})) \leq 2\epsilon$ with probability at least $1 - \frac{2}{100}$ over the choice of $h$. This proves *uniqueness*. $\qquad\square$

---

[13]If the sets were not disjoint then $\mathbf{B}$ could see it as suspicious because $f$ makes mistakes on all of $A_w$.

