# OpenReview forum: "The Good, the Bad and the Ugly: Watermarks, Transferable Attacks and Adversarial Defenses"
_ICLR.cc/2025/Conference — Submitted to ICLR 2025_

### Official Review · Reviewer_fUbR · 2024-10-20

**Soundness:** 3
**Presentation:** 3
**Contribution:** 3
**Rating:** 6
**Confidence:** 4

**Summary:**

The paper gives formal definitions of “watermarks” through backdoor (not for LLM’s output, but rather for models themselves) and “adversarial robustness” and “transferable attacks” in their own way. Meaning that the definitions do not necessarily match what is the common usual way of defining them, but the definitions make sense in their own way.

Then, the paper observes that these three notions are complementary for a “learning task”. A task is modeled using a distribution D (on instances) and a function h (to label them) and differs from the method of using a family of h (as hypothesis class). In particular, the paper shows that for each learning task, at least one of the following holds: either we can watermark models that predict that task, or that we can resist backdoor, or that transferability attacks work.

Intuitively, the main result is proved by observing that a watermark through a backdoor aims to plant a backdoor and later use specific queries to detect it (using wrong answer) and this is exactly what a defense against backdoor wants to avoid. So the two notions are rather complementary. Once the paper aims to prove this formally, they show that a third case is also possible, which in their formalism is referred to as the transferability attack.

The paper then shows that “transferability” attacks could probably exist assuming fully homomorphic encryption, and that transferability attacks *require* one-way functions (they say PRG, but that is the same as OWFs), and hence it implies secret key crypto.

Finally, the paper claims that PAC learnable families (ie., those with bounded VC dimension) always can have adversarial robustness and “watermark against fast adversaries”.

**Strengths:**

A formalism of the intuition behind the duality of watermarks (for models through backdoor) and adversarial robustness is interesting.

Also, the paper realizes that formal definitions are needed for such results and takes an effort in that direction.

**Weaknesses:**

The new formalisms for the 3 notions of watermark, robustness and transferability need a lot more scrutiny and discussion. For example, there are limits on the time of the adversary that are needed to make these definitions non-vacuous, but these definitions are different from previously established definitions in this regard (e.g., about adversarial examples) and I see no real effort to compare them. (see my question below)


Also, due to the number of results in the main body, their proofs are deferred to the appendix, and perhaps the most exciting result (saying that bounded VC dimension means we can have adversarial robustness) is pushed to the appendix.

**Questions:**

Regarding the differences in definitions (for adversarially robust learning): please provide a detailed comparison between your time-limited adversary definitions and established definitions in adversarial examples literature, and highlight the key differences and justify your choices.

Related to the question above: are you claiming that any PAC learnable problem has adversarially robust learners? This seems to be a very important and technical problem, e.g., see:
https://proceedings.mlr.press/v99/montasser19a/montasser19a.pdf
https://arxiv.org/pdf/1806.01471
I guess the devil is in the details and differences in definitions, but that needs to be much more openly discussed.

Having a comparison of the definitions (yours and previous literature on adversarial examples and adversarially robust learning) partially addresses this issue, but please also make an explicit comparison with the results of the papers above (particularly, the impossibility result of the first paper and how it does not contradict yours).

---

> ### Author Response · Authors · 2024-11-24
>
> ## Response to Reviewer fUbR
>
> We thank the reviewer for their comments relating to definitions of adversarial defenses and robust learnability. Below, we address the specific points raised.
>
> ---
>
> ### On Our Adversarial Defense Definition
>
> We would first like to clarify that the model of Adversarial Defense we work with is similar to models where abstaining from responding is allowed, i.e., Selective Classification. To this end, our result shows the existence of a defense for tasks with bounded VC-dimension, using the algorithm of [Goldwasser et al., 2020] as a subroutine. This ensures that there is no restriction on the model of attack of the adversaries.
>
> Furthermore, finding adversarial examples and defending against adversaries are problems inherently limited by computation ([Garg et al., 2020] and [Bubeck et al., 2019]). Hence, we believe it is natural to model adversaries as computationally limited entities.
>
> Importantly, our definition also allows adversaries that are unbounded. For instance, we show the existence of an adversarial defense in the case of learning tasks with bounded VC-dimension, even when attackers are computationally unbounded.
>
> ---
>
> ### On the Applicability of Lower Bounds of [Montasser et al., 2019]
>
> In our framework, the function output by the defense algorithm may differ from the hypothesis class of the target labeling function. This means our approach falls under the improper learning framework, and thus we believe there is no contradiction with the lower bounds presented in [Montasser et al., 2019].
>
> ---
>
> ### Additional Notes
>
> We hope that our responses clarify the concerns raised. If there are any remaining questions or points requiring further elaboration, we kindly ask the reviewer to let us know. We greatly appreciate your constructive feedback and its contribution to improving the paper.
>
> ---
>
> ### References
>
> - Bubeck, Sébastien, et al. *"Adversarial examples from computational constraints."* International Conference on Machine Learning. PMLR, 2019.
> - Garg, Sanjam, et al. *"Adversarially Robust Learning Could Leverage Computational Hardness."* Algorithmic Learning Theory. PMLR, 2020.
> - Montasser, Omar, Steve Hanneke, and Nathan Srebro. *"VC classes are adversarially robustly learnable, but only improperly."* Conference on Learning Theory. PMLR, 2019.

---

> > ### Comment · Reviewer_fUbR · 2024-11-26
> > **Re:**
> >
> > Thanks for your comments. I don't think the comparison with Montasser et al., 2019 (and the very rich line of work on provable defenses against adversarial examples) is still anywhere close to being complete. But I do see your point that the issue of being proper vs improper is one difference. I encourage the authors to add better comparisons with existing lines of work.
> >
> > Having said that, I still remain mildly positive about the paper and believe after taking into account the provided feedback, it provides a useful insight that is important to be formalized.

---

> > > ### Author Response · Authors · 2024-12-02
> > >
> > > We appreciate the reviewer’s follow-up comment and the suggestion to expand the comparisons with Montasser et al., 2019, and related work on provable defenses against adversarial examples. We recognize the importance of providing a more detailed discussion, and we will incorporate this in the final version of the paper to ensure the comparisons are thorough and clear.
> > >
> > > We are glad the reviewer finds the paper to provide useful insights and formalizations. Your feedback has been valuable in refining our work, and we will strive to address these points comprehensively.

---

### Official Review · Reviewer_7Ahn · 2024-11-01

**Soundness:** 2
**Presentation:** 2
**Contribution:** 3
**Rating:** 6
**Confidence:** 4

**Summary:**

The authors provably identify the following trichotomy - for every learnable task there is an adversarial defence and/or a watermarking scheme, while for learning tasks which have associated cryptographic hardness results, there is a transferable attack.

**Strengths:**

The most important contribution of this manuscript is the proof that the notions adversarial robustness and watermarking schemes is complementary to the notion of cryptographically hard learning schemes. The authors use a lot of existing results across various fields creatively, to arrive at this result, which makes the technical part of the paper interesting in its own right.

**Weaknesses:**

My main concern with the paper is that the definitions (especially the two player games) is too carefully constructed to be readily used in conjunction with existing results from game theory, cryptography, and learning theory. There is lack of justification / discussion on several fronts, which should be addressed for the paper to be useful to the community at large.

A secondary but related weakness is the lack of a technical discussion section. A detailed overview of proof techniques section is much required. I have detailed a list of my questions in the next section.

A better related works section is also warranted. For example, there are certain confusions arising in the discussion containing the comparison with the Christiano et al. (2024) paper. See the questions section.

*Note to the authors:* Regardless of the acceptance results at this conference, I believe the authors should prepare a longer version of this manuscript and submit it to a journal like TMLR. It would be of immense value to the community.


--------
After the rebuttal period, I have decided to raise my score to indicate my positive opinion of the paper.

*Note:* At this point I cannot justify raising the score any higher due to numerous missing definitions and discussions (detailed in the reviews by myself and other reviewers, and mostly agreed upon by the authors) that should have been included in a theoretical paper in the first place.

**Questions:**

# Main points to be addressed

1. In section 3.1 (line 202), why do we need to define $\perp$, given the fact that $x\sim D$ and $\text{dom}(h)=\text{supp}(D)$? It seems to me that we are never really using the fact (at least in the vicinity of said definition) that $h$ is a partial function? This is causing unneeded confusion at this point. This subtlety can be separately introduced in appendix F and H, where it is actually required.


2. The definition of indistinguishability which forms a cornerstone of most of the concepts in this paper is both _informal_ and _incomplete_.  The authors define the distinguishing game, but abruptly end the definition with $\text{Pr}\left[A\text{ wins}\right]$.
     - More concretely, the authors should explicitly state what the probability in line 215 is over, and connect the winning probability to the distinguishing game?


3. In definitions 5,6, and 7, it seems to be a straightforward requirement that $t<T$. The authors should include this in the paper.

4. Line 150 onwards: I request the authors for certain clarifications:
   - "_A major difference between our work and that of Christiano et al. (2024), is that in their approach, the attacker chooses the distribution, whereas we keep the distribution fixed._" Does this not imply that the related work proves a *stronger result*? (Note: this does not influence my score. I simply wish to understand this point in some more detail.)

   - "_A second major difference is that our main result holds for all learning tasks, while the contributions of Christiano et al. (2024) hold for restricted classes only._" Earlier, it was stated that the related work "_show(s) an equivalence between their notion of defendability (in a computationally unbounded setting) and PAC learnability._" So what do the authors mean by *restricted classes* in the work of Christiano et al. (2024)?

5. The notions of zero-sum games and Nash equilibria are absent from the main paper. This makes the proof sketch of Theorem 1 highly unreadable without jumping back and forth from appendix C, which defeats the whole purpose of having a simplified main theorem and a proof sketch in the first place.
   - Why do we need to restrict ourselves to zero-sum games instead of focusing on more general interactive protocols? Is it simply because the proof framework demands a certain kind of assumption? In this case, such a setup should be clearly mentioned as an assumption (which is still fine).
   - The authors should also address why they have only considered Nash equilibria instead of in their setup, if not for other reasons, for the sake pedagogy alone.

6. Succinctly representable circuits are not well-motivated in Appendix C. One the referenced papers uses the notion of Stackelberg equilibrium which is different from Nash equilibrium used in this paper. There needs to be a longer/better discussion on this point.

7. It seems to be that FHE is superfluous in the proof of Theorem 2. Any reasonable encryption task seems to suffice for the purposes of the proof.
   - On this note, I have the following question for the authors: Consider the Learning with Errors problem (Regev, 2005), where the learner has to figure out if samples are drawn from the LWE distribution or the uniform distribution. The adversary in LWE can be taken as Alice, while the learner, i.e., Bob has to figure out if the samples were drawn from uniform or the LWE distribution.

## Typos
  - Line 1599: "For instance in ? the existence of PRGs secure against time and space bounded adversaries was considered." *Missing reference*.

---

> ### Author Response · Authors · 2024-11-24
>
> ## Response to Reviewer 7Ahn
> We thank the reviewer for their thoughtful and detailed feedback, which has helped us identify areas for improvement and clarify our work. Below, we address the comments raised.
>
> ### 1. On the Necessity of the Definition
> The definition is necessary since samples that are adversarial could be outside the $supp(D)$. We will make the definition clearer in the main draft.
>
> ---
>
> ### 2. Advantage and Indistinguishability
>
> We clarify the concepts of *advantage* and *indistinguishability* here, through a game between a sender and a distinguisher.
>
> For an algorithm $\mathcal{A}$ (also known as the distinguisher) and two distributions $\mathcal{D}_0, \mathcal{D}_1$, consider the following game between a sender and the distinguisher:
>
> 1. The sender samples a bit $b \sim U(\{0,1\})$ and then draws a random sample $\mathbf{x} \sim \mathcal{D}_b$.
> 2. $\mathcal{A}$ receives $\mathbf{x}$ and outputs $\hat{b} := \mathcal{A}(\mathbf{x}) \in \{0,1\}$. $\mathcal{A}$ wins if $\hat{b} = b$.
>
> We say that $\delta \in (0, \frac{1}{2})$ is the *advantage* of $\mathcal{A}$ for *distinguishing* $\mathcal{D}_0$ from $\mathcal{D}_1$ if:
>
> $$
> P_{b \sim U(\{0,1\}), \mathbf{x} \sim \mathcal{D}_b}[\mathcal{A}(\mathbf{x}) = b] = \frac{1}{2} + \delta\;,
> $$
>
> For a class of algorithms, we say that the two distributions $\mathcal{D}_0$ and $\mathcal{D}_1$ are $\delta$-*indistinguishable* if for any algorithm in the class, its advantage is at most $\delta$.
>
> ---
>
> ### 3. Generality of Definitions
> The definitions are general, and it could be that the protocols are instantiated with $t \geq T$. For instance, in our Transferable Attack example (Theorem 2), we have $t > T$.
>
> ---
>
> ### 4. Comparability of Definitions
>
> In general, these definitions are incomparable:
>
> - You are correct that if the attacker can choose a distribution, it makes defendability (in the sense of Christiano et al.) harder compared to a fixed distribution (as in our model). On the other hand, the backdoor $x^*$ is sampled $\sim \mathcal{D}$ in Christiano et al., i.e., the attacker has no influence over it, while in our model, the attacker can choose $x$'s. In this aspect, defendability is easier in Christiano's model.
>
> - It is interesting to investigate further that both we and Christiano showed a Defense/defendability for bounded VC-dimension classes, albeit with a priori incomparable definitions.
>
> We acknowledge that we did not phrase the comparison perfectly. Our intention was to emphasize the following differences in the respective frameworks:
>
> 1. Christiano et al. aim at a taxonomy of representation classes only (independent of $\mathcal{D}$), while our Theorem 1 gives a taxonomy of pairs $(\mathcal{D}, h)$.
> 2. This allows us to model scenarios where $\mathcal{D}$ is chosen by nature and not just in the worst-case setting.
>
> ---
>
> ### 5. Use of Zero-Sum Games
>
> Using zero-sum games for us is a proof strategy and not a modeling choice, as our proof examines how attacks on watermarks can be turned into defenses and vice versa. Hence, zero-sum games are a natural choice. Moreover, zero-sum games are cleaner in the sense that the value of the game is unique, which prevents issues about equilibrium selection.
>
> Regarding other equilibrium notions, such as Stackelberg equilibria, in two-player zero-sum games, the set of Stackelberg equilibria and Nash equilibria is the same. Additionally, one can show that coarse correlated equilibria (CCEs) and Nash equilibria are nearly equivalent for two-player zero-sum games.
>
> We will add more background on two-player zero-sum games and Nash equilibria in the main paper for the final version.
>
> ---
>
> ### 6. Succinct Representation
>
> Please see the Succinct Representation paragraph in our response to Reviewer EjNG.
>
> ---
>
> ### 7. Transferable Attack and LWE Clarification
>
> We are not entirely sure how to construct the Transferable Attack example without using FHE, as the learning procedure requires computation on encrypted data. We would be open to learning about other methods if they simplify the construction.
>
> Regarding the reviewer's question about the LWE problem, we note the following:
>
> 1. LWE is used to build FHE, so it is unclear if the proposed construction uses a weaker assumption.
> 2. If we understand your point correctly, the two distributions you define, i.e., LWE and uniform, are indeed indistinguishable as required by undetectability in Definition 3. However, we do not see how transferability is satisfied or how the ground truth is defined in the first place.
>
>
> ### Additional Notes
>
> We kindly ask you to confirm whether our responses have resolved your concerns and clarified the issues raised. If there are any remaining questions or points requiring further elaboration, please let us know. Your thoughtful feedback is valuable in helping us improve our work.

---

> > ### Comment · Reviewer_7Ahn · 2024-11-24
> >
> > > LWE is used to build FHE, so it is unclear if the proposed construction uses a weaker assumption.
> >
> > I don't think I mentioned the need for a weaker assumption in the review. Showing that LWE suffices instead of FHE simplifies a lot of the prerequisites in that section.
> >
> > > . . . . we do not see how transferability is satisfied or how the ground truth is defined in the first place.
> >
> > I invite the authors to think about the hardness implications for LWE if transferability is satisfied. The ground truth is simply the label computed by the LWE concept.
> >
> > -----
> > ## Final Comments
> >
> > I am mostly satisfied with the responses. The authors should include all missing discussions and definitions in the revised manuscript. I am changing my score to reflect my positive opinion on the paper.

---

> > > ### Author Response · Authors · 2024-11-25
> > >
> > > We thank the reviewer for this insightful comment.
> > >
> > > If we understand the comment correctly,
> > > the reviewer's proposal of an LWE-based Transferable Attack is such that it is impossible to learn a low-error classifier in the given time bound, and so transferability is automatically satisfied.
> > > The undetectability follows from the properties of LWE.
> > > In such a regime, an even more extreme construction is possible where one considers any hard-to-learn task and makes $\mathbf{A}$ set $\mathbf{x} = \mathcal{D}^q$.
> > > This makes the undetectability trivial as the two distributions are identical, and transferability follows from the hardness of learning.
> > >
> > > In our paper, the regime of interest is always that at least one of the parties has enough time to learn $f$ of error $\textbf{err}(f) \leq \epsilon$ (see the requirement of efficient learnability in Theorems 1 and 3, and the last sentence of Theorem 2, which guarantees our Transferable Attack to be in this regime of interest).
> > > To our understanding, the reviewer's LWE-based construction does not satisfy this requirement as learning is hard.
> > >
> > > We will make the assumption about the regime of interest more explicit in the final version of the manuscript and also add the missing discussions and definitions as per the reviewer's comment.

---

### Official Review · Reviewer_tRxz · 2024-11-01

**Soundness:** 3
**Presentation:** 3
**Contribution:** 2
**Rating:** 5
**Confidence:** 2

**Summary:**

This paper explores the relationship between backdoor-based watermarks and adversarial defenses in machine learning.  These concepts are formalized as interactive protocols between two players and proved that for almost every discriminative learning task, at least one of the two exists. The main contribution is the identification of a third, counterintuitive option: a transferable attack.  This term describes an algorithm capable of generating queries that are indistinguishable from the data distribution, yet can deceive even the most effective defenses.  The authors demonstrated the necessity of a transferable attack using homomorphic encryption and proved that any task susceptible to such an attack implies a cryptographic primitive.

**Strengths:**

- Formalization of the relationship between backdoor-based watermarks and adversarial defenses is useful.

**Weaknesses:**

- I take umbrage with the following claim: “Along the way to proving the main result, we identify a potential reason why this fact was not discovered earlier.”. There are multiple prior works that have investigated the trade-off between adversarial robustness and backdoors/watermarks [Weng et al,  Sun et al, Gao et al, Niu et al., Fowl et al., related work in Tao et al. is a good summary]. Although most of these papers are more empirical, this paper completely ignores an entire line of work. The paper primarily focuses on theoretical results without providing clear guidance on how these results can be translated into practical applications, and I find it difficult to assess if this paper is saying anything profound beyond what has already been discussed in the referenced papers. This is my main concern. Some suggestions:
    * (1) Include a detailed discussion of how their theoretical results relate to or extend the empirical findings in the papers you cited.
    * (2) Explicitly state what novel insights their work provides beyond the existing literature.
    * (3) Add a section on potential practical applications or implications of their theoretical results.

- In Definition 2.3. Why is the coefficient before epsilon, 7?

- The paper primarily deals with discriminative learning tasks, like classification. These tasks assume a clear relationship between input data (e.g., images) and distinct output labels (e.g., "cat" or "dog").  How can the trade-offs be captured for generative models?


[Weng et al] Weng, Cheng-Hsin, Yan-Ting Lee, and Shan-Hung Brandon Wu. "On the trade-off between adversarial and backdoor robustness." Advances in Neural Information Processing Systems 33 (2020): 11973-11983.

[Sun et al] Sun, Mingjie, Siddhant Agarwal, and J. Zico Kolter. "Poisoned classifiers are not only backdoored, they are fundamentally broken." arXiv preprint arXiv:2010.09080 (2020).

[Gao et al.] https://openreview.net/forum?id=nG4DkcHDw_

[Niu et al.] Niu, Zhenxing, et al. "Towards unified robustness against both backdoor and adversarial attacks." IEEE transactions on pattern analysis and machine intelligence (2024).

[Fowl et al.] Fowl, Liam, et al. "Adversarial examples make strong poisons." Advances in Neural Information Processing Systems 34 (2021): 30339-30351.

[Tao et al.] Tao, Lue, et al. "Better safe than sorry: Preventing delusive adversaries with adversarial training." Advances in Neural Information Processing Systems 34 (2021): 16209-16225.

**Questions:**

See above.

---

> ### Author Response · Authors · 2024-11-24
>
> ## Response to Reviewer tRxz
>
> We thank Reviewer tRxz for their time and effort in providing detailed and constructive feedback on our work. Below, we address each of the points raised, organized into sections for clarity.
>
> ### 1. Prior Work on Trade-offs
>
> #### Reviewer Comment:
> *“There are multiple prior works that have investigated the trade-off between adversarial robustness and backdoors/watermarks... Although most of these papers are more empirical, this paper completely ignores an entire line of work.”*
>
> #### Response:
> We appreciate the reviewer highlighting the importance of prior work on adversarial robustness and backdoor trade-offs. While our work primarily focuses on a theoretical framework, we agree that an explicit discussion of how our results relate to existing empirical studies would enhance the paper. To this end:
>
> - In Appendix A.2 (Adversarial Defense), we will add a discussion about our theoretical framework and prior empirical studies, such as [Weng et al., Sun et al., Niu et al., Fowl et al., Tao et al.]. While these works explore empirical trade-offs, our framework formalizes these relationships through interactive proof systems and cryptographic primitives.
> - We will emphasize in Section 2 (Related Work) that our framework introduces the concept of transferable attacks to this trade-off, not previously explored in these works, as a theoretical tool to prove the simultaneous failure of adversarial robustness and backdoor defenses.
>
> We will also clarify the phrasing of the statement: *“Along the way to proving the main result, we identify a potential reason why this fact was not discovered earlier”* (Section 1.1.) to avoid ambiguity. It will explicitly reference the theoretical conditions underpinning our main result, rather than implying the absence of prior work.
>
> ---
>
> #### Reviewer Comment:
> *“Explicitly state what novel insights their work provides beyond the existing literature.”*
>
> #### Response:
> In Section 1.1, we explicitly outline our contributions, which we summarize as follows:
>
> 1. **We prove that for every learning task, at least one of the following must exist: a backdoor-based watermark, an adversarial defense, or a transferable attack.**
>    This result extends the empirical observations in prior work by providing a formal theoretical framework that guarantees these outcomes for all learning tasks.
>
> 2. **We introduce the concept of a transferable attack, a third and previously unexplored option in this trade-off.**
>    Transferable attacks arise for tasks where neither watermarks nor adversarial defenses can exist. This counterintuitive result broadens the theoretical understanding of trade-offs in robustness and backdoor-based watermarks by identifying scenarios that were overlooked in prior empirical studies.
>
> 3. **We establish a connection between transferable attacks and cryptographic primitives.**
>    We show that the existence of a transferable attack for a learning task implies the presence of certain cryptographic properties, emphasizing the computational complexity inherent in these tasks.
>
> ---
>
> #### Reviewer Comment:
> *“Add a section on potential practical applications or implications of their theoretical results.”*
>
> #### Response:
> We thank the reviewer for this suggestion. The practical implications of our theoretical results are discussed in Section 7 (*Implications for AI Safety*). In this section, we conjecture that adversarial defenses will exist for safety-critical discriminative learning tasks and provide three supporting arguments based on our theoretical findings, including Theorem 1 and the connection between transferable attacks and cryptographic primitives (Theorem 8). These insights directly contribute to understanding the robustness and security of AI systems in safety-critical regimes.
>
> ---
>
> ### 2. Coefficient Before Epsilon in Definition 2.3
>
> #### Reviewer Comment:
> *“Why is the coefficient before epsilon, 7?”*
>
> #### Response:
> We thank the reviewer for raising this question.
>
> The coefficient $7$ in Definition 2.3 is not a special constant. Intuitively, it comes from a type of "triangle inequality" in the proof of Theorem 1, where one needs to account for multiple sources of error (e.g., $2\epsilon + 2\epsilon + 2\epsilon < 7\epsilon$). We will clarify this reasoning in the final version to improve accessibility by adding an explanation to Appendix C (Main Theorem).
>
> *Note: The response continues in the next section.*

---

> > ### Author Response · Authors · 2024-11-24
> >
> > *Continuation of the response from the previous section.*
> > ### 3. Applicability to Generative Models
> >
> > #### Reviewer Comment:
> > *“How can the trade-offs be captured for generative models?”*
> >
> > #### Response:
> > We address the applicability of our framework to generative models in Appendix D (*Beyond Classification*), where we discuss how the dynamics of generation and verification differ fundamentally from classification tasks and explore the potential extension of our results to generative settings. To improve visibility, we will include a reference to Appendix D in Section 2, immediately following the discussion of backdoor-based watermarks in Pre-trained Language Models (PLMs).
> >
> > ---
> >
> > ### Additional Notes
> >
> > We appreciate the reviewer's comments regarding their confidence level and understand the challenges in assessing a theoretically dense submission. We aim to improve clarity throughout the manuscript in the final version.
> >
> > We hope these responses and planned revisions address the reviewer's concerns. We are grateful for your constructive feedback, which provides valuable improvements to our work. We kindly ask you to confirm if our proposed revisions resolve your concerns or if further clarification is needed.

---

### Official Review · Reviewer_EjNG · 2024-11-04

**Soundness:** 1
**Presentation:** 2
**Contribution:** 2
**Rating:** 3
**Confidence:** 3

**Summary:**

This paper investigates the relationship between watermarks (planted in trained ML models) and adversarial defenses for noiseless classification tasks over a finite set $\mathcal{X}$. For clarity, let us focus on *binary* classification tasks. Here, a learning task (or equivalently, full data distribution) can be represented by a pair $(D, f^*)$, where $D$ is the marginal distribution over $\mathcal{X}$ and $f^*: \mathcal{X} \to \\{0,1\\}$ is the true labeling function.

For any given learning task $(D, f^*)$, consider an interactive protocol in which Alice (ML service provider) interacts with Bob (client). As a service provider, Alice trains a classifier $f: \mathcal{X} \to \\{0,1\\}$ which achieves $\epsilon$-error on $D$, and sends it to Bob. However, Alice is motivated to secretly plant a “watermark” into her trained classifier $f: \mathcal{X} \to \\{0,1\\}$ by making it susceptible to pre-designed adversarial examples. Bob, on the other hand, is motivated to neutralize any backdoors in the classifier f he received from Alice.

The opposing objectives of Alice and Bob in this framework can be formulated as a *zero-sum two-player* game. Furthermore, by modeling Alice and Bob to be circuit classes of fixed size, the pure strategy space for both Alice and Bob become finite, with explicit bounds on their cardinalities. This setup allows previous results on approximate equilibria [Lipton and Young, 1994] to be applied. Using the zero-sum two-player game formulation, the authors show that any “efficiently learnable” classification task $(D, f^*)$ falls into at least one of the following three cases:

1. **Watermarking.** There exists a watermarking scheme for Alice can compute a classifier f and sequence of adversarial (randomized) queries $x_1, …, x_q$ such that for any circuit (”Bob”) whose size is significantly smaller than hers (i.e., y computed by any such small circuit incurs $\mathrm{err}(x, y) \ge 2\epsilon$) the watermark is *unremovable* and the distribution of her queries $x_1, …, x_q$ is indistinguishable from $D$.
2. **Adversarial Defense.** There exists a watermark neutralizing (i.e., adversarial defense) scheme for Bob such that either Alice’s queries $x_1,…,x_q$ are non-adversarial (the avg loss of $f$ on $x_1, …, x_q$ is $\epsilon$-small) or the distribution of Alice’s queries $x_1, …, x_q$ and $D^q$ are distinguishable by small circuits.
3. **Transfer Attack.** A third possibility not covered by the previous two cases, which has left me confused. Please refer to the Questions section for further details.

**Strengths:**

The paper attempts to formalize an intriguing relationship between two phenomena recently studied in machine learning: watermarking (i.e., planting undetectable backdoors [Goldwasser et al., 2022]) and adversarial defense [Cohen et al., 2019]. A watermark for a classifier attempts to hide specific signatures in its error patterns, while an adversarial defense attempts to maintain performance of a given untrusted classifier f across distributions that are “close to” D, which can be formalized via a weakened notion of statistical distance. Intuitively, there is tension between these two objectives, which the authors attempt to formalize as a zero sum game. Addressing these natural questions would be of wide interest to the ML community.

**Weaknesses:**

The main weakness of this paper lies in the **lack of clarity and precision in its definitions and framework**, which significantly undermine the credibility of any theorems that follow. The presentation of key definitions and interactive protocols are “hand-wavy”, leaving substantial ambiguity in how the results should be interpreted and applied. This vagueness makes it difficult to assess the validity of the theoretical claims and fully appreciate the significance of the results.

While the authors give more formal specifications in the Appendix (especially, Appendix B), significant gaps still remain to be filled. In addition, the appendix should provide further technical details after the basic setup and key insights have been presented in the main text, rather than serving as a teaser for readers left confused by the unclear presentation in the body of the work.

One significant issue is that the modeling of Alice and Bob with size-bounded circuit classes seems to fail a basic type check. In the interactive protocols, Alice and Bob face different tasks that involve different input and output spaces. For instance, in a watermarking scheme for binary classification, Alice is expected to output a representation of a classifier $f: \mathcal{X} \to \\{0,1\\}$ along with queries $ x_1,\ldots,x_q \in \mathcal{X}$ (a separate issue here is Alice's inputs are not specified and the dependence on the input domain’s cardinality doesn't appear anywhere in the quantitative results, which raises concerns). On the other hand, Bob’s inputs are sequences $x_1, \ldots, x_q$ and needs to output $y \in \\{0,1\\}^q$. Without further clarification, it’s unclear how a size $s$ circuit for Alice compares to a size $s$ circuit for Bob since even the input and output domains do not match. This is one of several issues with the paper's framework that, collectively, call into question the overall rigor and applicability of the approach. Please refer to the **Questions** section for additional issues.

Moreover, the paper **incorrectly applies previous results from cryptography**, which indicates a lack of understanding of the field. In particular, the interpretation of the results based on [Goldreich, 1990] in Section 5.2 is incorrect. Goldreich’s result applies to an *ensemble* of random variables, i.e., a *sequence* of distributions, whereas the EFID pairs the authors define in Section 5.2.1 are particular instances of distributions. Moreover, the ensembles used by Goldreich are *uniformly constructible*, meaning that there exists a single Turing machine generating random samples from X_n given the input 1^n. This contrasts with the non-uniform circuits used in this work. Given this misunderstanding, the title of Section 5,  *Transferable Attacks are “Equivalent” to Cryptography* is misleading and unnecessarily provocative. Even if Goldreich’s equivalence result could be applied here (which I find unlikely), the only concrete implication for cryptography is the existence of pseudorandom generators (PRGs), which are basic cryptographic primitives but do not represent the entire research field.

In addition, the restriction to succinct circuits feels somewhat ad hoc, seemingly added specifically to prevent Alice and Bob from hardcoding outputs. It seems likely that the approximate equilibria results (Theorem 1) would still hold without the succinctness assumption, albeit with different bounds, as the key requirement is simply that the pure strategy space remains finite with explicitly known bounds on its cardinality. This raises concerns about the integrity of the formulation, with the succinctness restriction serving more as a workaround than an integral component of the setup.

Overall, the paper would benefit greatly from prioritizing clarity over excessive generality, focusing on straightforward, concrete setups and presenting mathematical results clearly and precisely, without the informal remarks.

**Editorial comments**
- (Abstract) The phrase "almost every" learning task is misleading. Terms like “almost every” carry strong connotations in measure theory. The mere fact that a mathematical object is “irregular” or "very complex" does not imply that it is rare (e.g., from a measure-theoretic perspective). For instance, with respect to the uniform measure on [0,1], “almost every” real number in the unit interval is uncomputable.
- (Line 214) Advantage should be an equal sign.
- (Line 243) The unremovability condition, as stated, is clearly incorrect. Bob can simply respond with random y and the realized error can be 0 with small but non-zero probability. Even if this is intended as an informal simplification of the definitions in Appendix B, it should not be so obviously wrong.
- (Line 248) The term "defender" is used inconsistently alongside other terms like "player", "prover", and "Bob". It would be better to choose a single term for the recurring entities and use it consistently throughout the paper.

**Questions:**

Many issues appear to stem from portraying Alice (and Bob) in an “active” or “anthropomorphic” manner, as if she dynamically performs tasks, which is inconsistent with the static nature of non-adaptive computation models like circuits. Circuits compute a fixed function based purely on their inputs.

**Formal specification of Alice and Bob**
1. What mathematical objects represent Alice and Bob? If they are fixed-size circuits, then what is the input and output space of these circuits? How are the input spaces structured, and how are random source bits, which are necessary for a circuit to output a random variable, represented within these spaces? Do they count towards the size of the circuit?
2. Circuits themselves are not formally defined anywhere in the text, so it’s unclear what the authors mean by a “size $s$ circuit” in mathematically. The closest reference is [Appendix B, Definition 4] which defines "succinct circuits" but leaves "width" and "depth" undefined. For instance, what would be the width and depth of a circuit shaped like a pyramid? In addition, are input gates counted as part of the circuit size?

**Specification of the interactive protocol (Section 3.2)**
1. How does the interactive protocol proceed? Does Alice send $(f, x_1,…, x_q)$ to Bob in a single round? Are there multiple rounds?
2. If there are no assumptions on the hypothesis class, how does Alice represent her classifier $f: \mathcal{X} \to \\{0,1\\}$? The only reasonable choice in the absence of structural assumptions on the labeling function seems to be the truth table, i.e., an element of $\\{0,1\\}^{\mathcal{X}}$.
3. It seems weird to compare Alice’s budget $T_A$ and Bob’s budget $T_B$. Within the protocol, they have different inputs and different outputs. Alice must output $(f, x_1, …, x_q)$ which could be potentially much longer than Bob’s $q$ bits to represent $(y_1,…,y_q)$.
4. In the *Transfer Attack* setting, what access does Bob have to the learning task $(D, f^*)$? Figure 4 suggests that Bob is only provided with Alice’s queries. Is it reasonable to expect Bob to achieve low error on Alice’s queries without any access to the labeling function? Moreover, what is the significance of this setup if Bob is not given any information about the learning task itself?

**Other confusions**
1. In [Section 4] (also [Appendix C, Theorem 5]), what is meant by a “learning task $(D, f^*)$ is learned by a circuit up to error $\epsilon$”? How does a circuit solve a learning task? What are its inputs? What is the output? How is the output represented? What kind of “access” does a circuit have to the learning task $(D, f^*)$?

2. If $\mathcal{X} = [N]$, where $N=2^{100}$, $D$ is uniform over $[N]$, $f^* = 1$ everywhere, then what case does the learning task $(D, f^*)$ fall into? If the circuit class size bound for both Alice and Bob are small, say less than $10$, then Alice cannot even write down a description of a single sample from $\mathcal{X}$, which makes the *Watermark* and *Adversarial Defense* schemes ill-defined. On the other hand, the learning task is technically "learnable" since a constant circuit that is not connected to any input gates and outputs $1$ realizes $f^*$.

---

> ### Author Response · Authors · 2024-11-24
>
> ## Response to Reviewer EjNG
> We sincerely thank the reviewer EjNG for their thoughtful and constructive feedback. Below, we address each of the concerns raised and provide clarifications where necessary.
>
> When writing this paper, we made a deliberate design choice to present the results in a manner that is accessible to a broad ML audience. Most research fields have an implicitly agreed-upon level of formality, which is not yet established for topics such as watermarks, adversarial defenses, and AI security more broadly. In our paper, we aim to bridge practical and theoretical perspectives.
>
> To this end, our definitions are designed to model very general threat scenarios, ensuring practical relevance. At the same time, we strive to provide formal and complete proofs to uphold theoretical rigor. Balancing these objectives is inherently challenging, and as you noted, the desired generality can make the presentation more complex.
>
> With this in mind, we fully agree that our paper is a theory-driven work and requires rigorous treatment. In some aspects, particularly in the main body, we may have oversimplified certain definitions and proofs. We will address your detailed comments below and aim to convince you that all notions can indeed be made formal. Naturally, we will incorporate your feedback into the final version of the paper.
>
> ### On Modeling
>
> #### Size Parameter
> We believe that a big part of the confusion stems from the fact that the size parameter is not present in our modeling. In computational complexity, one studies models of computations like DFAs, Turing Machines, and circuits. When studying DFAs, the size parameter is not present as the same automaton is used for all input sizes.
>
> In contrast, when analyzing the running time of Turing Machines or the sizes of circuits, one often evaluates complexity in terms of input length. Introducing a size parameter enables discussions about complexity classes like linear, polynomial, or exponential, without being constrained by constants. However, once an algorithm is instantiated in practice, the input size becomes fixed, and these constants start to matter.
>
> In our work, both the definitions and the main result (Theorem 1) are framed without a size parameter, focusing instead on a fixed learning task with a fixed-sized representation of samples. This choice stems from the observation that for the most general learning tasks, there is no clear, universally appropriate size parameter. For instance, should ImageNet be parameterized by the number of pixels? Such ambiguity underscores our decision to avoid introducing a size parameter unnecessarily.
>
> #### Succinctness
> You are correct in noticing that the succinct representation prevents hardcoding of the function, but our motivation stems from how learning happens in practice. More precisely, in our framework, the learning task is a fixed distribution and target labeling function. By using succinct representation, we enforce the lack of knowledge about the task, which prevents hardcoding $f$.
>
> In practice, we are often given fixed learning tasks, and constraints are imposed on the learning algorithm (e.g., gradient descent on a specific architecture or a time bound). As we mentioned in the paper, our model covers the case of neural networks because succinct circuits can represent gradient descent on NNs. In summary, both in practice and in our framework, it is impossible to hardcode the true labeling functions for non-trivial learning tasks, which are the ones we care about.
>
> ### Cryptography
>
> In Section 5.1 (Theorem 2, also see Theorem 7 with all parameters), we use the size parameter. This is because we leverage existing cryptographic tools, such as FHE, where efficiency/security is defined as polynomial/polynomial indistinguishability. Our construction considers a family of distributions (that define the learning task) parameterized by $\lambda$ (see Theorem 7). However, in practice, the parties receive samples only from one of these distributions.
>
> In Section 5.2, where we discuss EFID pairs, we do not use the size parameter. We acknowledge that this could lead to confusion and will emphasize this distinction more clearly. As you point out, the usual definition of EFID pairs considers ensembles and polynomial indistinguishability.
>
> We also appreciate your observation regarding the Goldreich construction of pseudorandom generators, which does not apply in our case due to the reliance on randomness during EFID pair generation. However, we believe a more careful adaptation of the construction might still yield a pseudorandom generator in our framework.
>
> In summary, we agree that the title of this section might be too provocative, as we demonstrate the existence of an EFID pair rather than addressing all cryptography. However, we stand by the justification that the pair of distributions we define can be considered a *type* of EFID pair.
>
> *Note: The response continues in the next section.*

---

> > ### Author Response · Authors · 2024-11-24
> >
> > *Continuation of the response from the previous section.*
> >
> > ### Circuits
> >
> > Definition 4 implicitly implies that we are interested in layered circuits. This is because circuit $C$ specifies to which gates of the previous layer a given gate is connected. We defined the size of a circuit as $\text{size}(C) := w \cdot d$, where $w$ is the width of the circuit and $d$ its depth. This implies that a circuit in the shape of a pyramid will have a size equal to its depth times the width of its base, i.e., we count empty spaces toward the size.
> >
> > ### Specification of the Interactive Protocol (Section 3.2)
> >
> > 1. For the case of a watermark, $f, x_1, \dots, x_q$ are sent in one round.
> > 2. Formally, Alice and Bob should agree on a representation of $f$ before the interaction starts. This representation could take the form of a truth table, a specific neural network architecture, or any hypothesis class. Importantly, Theorem 1 remains unchanged, as it guarantees that for every representation class, one of the three schemes exists.
> > 3. It is true that Alice's input (for a watermark) can be much longer than Bob's. However, this does not invalidate Theorem 1. Guaranteeing a watermark when $T_B \approx T_A$ is only harder than when $T_B \ll T_A$.
> > 4. As stated in Line 205, in all protocols, all parties (including Bob) have access to labeled i.i.d. samples from $\mathcal{D}$. With this clarification in mind, we encourage you to revisit Section 5.1 and Theorem 2, which we believe provides a surprising result.
> >
> > ### Other Confusions
> >
> > 1. All parties (and circuits) have access to i.i.d. labeled samples from $\mathcal{D}$. Thus, the inputs to the circuit that learns up to error $\epsilon$ are these labeled samples and potentially random bits.
> > 2. There should be a formal requirement that the respective circuits are large enough to represent at least $q$ (number of queries) elements of $\mathcal{X}$.
> >
> > ### Editorial Comments
> >
> > - **Abstract:** You are correct that the phrase "almost every" has measure-theoretic connotations. We will revise this phrasing in the final version.
> > - **Line 243:** The simplification might have gone too far. However, we still believe this description of unremovability is morally correct.
> >
> > ### Additional Notes
> >
> > We again thank you for your valuable thoughts and feedback. We kindly ask you to confirm whether our responses have resolved your concerns and clarified the issues raised. If there are any remaining questions or points that need to be further clarified, we would be very grateful if you would let us know.

---

> > > ### Comment · Reviewer_EjNG · 2024-11-26
> > >
> > > I thank the authors for engaging with my review and providing partial clarifications. However, I still find the framework and exposition problematic. Therefore, I maintain my initial rating of this work. I suggest starting with a simple, theoretically robust framework and then building on it with extensions, rather than proposing an overly general framework from the beginning.
> > >
> > > I reiterate two main issues, in response to the rebuttal: **lack of clarity and precision** and **notions of computational complexity without asymptotics or sequences**.
> > >
> > > **Lack of clarity and precision.** As the authors acknowledge in their rebuttal, this has already led to at least two significant issues, not to mention the potential confusion it could cause for readers. It has led to a significantly incorrect misinterpretation of their result in Section 5, which disqualifies at least half of the claim stated in the section title "Transferable Attacks are Equivalent to Cryptography". This is even more concerning since computational indistinguishability, the notion that was misunderstood by the authors, is a basic building block for most results in cryptography. The authors claim that
> > >
> > > > *However, we believe a more careful adaptation of the construction might still yield a pseudorandom generator in our framework.*
> > >
> > > I fail to see the relevance of this claim without accompanying proof or clear intuition to support it.
> > >
> > > The other issue is a trivial "counterexample" to their main theorem (Theorem 1), which I mentioned in **Other confusions** and arises directly from the underspecification of the framework. The authors proposed an ad hoc "bug fix" for this counterexample.
> > >
> > > > *There should be a formal requirement that the respective circuits are large enough to represent at least  $q$ (number of queries) elements of $\mathcal{X}$.*
> > >
> > > However, this fails to address the root cause of the problem, which is the insufficiently precise definition of the framework itself. Without further formalization, this "patch" does not resolve the underlying issues and does not convincingly show that more trivial counterexamples cannot be found.
> > >
> > > **Computational complexity without asymptotics or sequences.** The paper is largely about separations/impossibilities that arise from computational restrictions. However, in the main theorem statements, the time complexity $T$, is a given *number* rather than a function (or sequence) $T: \mathbb{N} \to \mathbb{N}$. Furthermore, the authors state that
> > >
> > > > *both the definitions and the main result (Theorem 1) are framed without a size parameter, focusing instead on a fixed learning task with a fixed-sized representation of samples.*
> > >
> > > This framing, devoid of any sequential notions, is questionable from a complexity perspective. Computational complexity is inherently an asymptotic notion (see e.g., Section 3.2 of Wigderson's *Mathematics and Computation* (2019)). This is why theorists often equate "time-efficiency" with "polynomial-time" or "polynomial-sized circuits"; it refers to how the time complexity *scales* with the size of the problem instances. For example, sorting 1000 numbers is clearly more difficult than sorting 2 numbers, but sorting is "efficient" because the time complexity scales as n log n for n numbers.
> > >
> > > **Secondary nitpicks.**
> > >
> > > > *To this end, our definitions are designed to model very general threat scenarios, ensuring practical relevance.*
> > >
> > > The aim for practical relevance seems to be at odds with the "truth table" representation of functions for Alice. A truth table representation scales with the *domain size*, which can easily exceed "the number of atoms in the universe" for moderate dimensions. For a theorist, the "practical" approach is to restrict the scope of one's results, which, in this case, would involve selecting a specific hypothesis class.
> > >
> > > > *For instance, should ImageNet be parameterized by the number of pixels? Such ambiguity underscores our decision to avoid introducing a size parameter unnecessarily.*
> > >
> > > The choice of the input size $n$ indexing is a design choice and different “time complexities” would arise from each choice. Yes, number of pixels is a valid choice. In fact, indexing images by the number of pixels is common practice in signal processing and high-dimensional statistics, as it highlights the “low-rankness” of images. For example, representation of natural images in the wavelet basis leads to sparse coefficients. Here, “sparse” is relative to the ambient dimension, which is the number of pixels.

---

> > > > ### Author Response · Authors · 2024-12-02
> > > >
> > > > We thank the reviewer for their thoughtful suggestion and engagement in the discussion.
> > > >
> > > > The reviewer proposes *“starting with a simple, theoretically robust framework and then building on it with extensions, rather than proposing an overly general framework from the beginning.”* In essence, our current setup and theory align with this principle. Specifically, Theorem 1 is presented for fixed circuit and input sizes, providing a foundational framework that is straightforward and theoretically robust. This serves as a natural starting point for understanding the problem.
> > > >
> > > > We acknowledge that, as the reviewer suggests, an alternative approach could involve considering a sequence of learning tasks indexed by a size parameter. In this case, Theorem 1 could be reformulated to assert that one of the three schemes (Watermark, Adversarial Defense, or Transferable Attack) occurs for infinitely many values of the size parameter. While this presentation might align more closely with the suggested approach, it would ultimately obfuscate the main contribution of the result. The proof of such a generalized theorem would still rely on independently applying Theorem 1 for each size parameter, adding unnecessary complexity to the setup without introducing new insights.
> > > >
> > > > In summary, we aimed to present the results as transparently as possible, avoiding additional complexity that could obscure the core contributions. By doing so, we believe our work already reflects the spirit of the reviewer’s request to prioritize simplicity and clarity in the foundational framework.
> > > >
> > > > ---
> > > >
> > > > ### Cryptography
> > > >
> > > > As mentioned, we do not currently have a proof to demonstrate that Transferable Attacks imply pseudorandom generators. However, consider the following distribution:
> > > > With probability $1/2$, sample $x \sim \mathcal{D}$ and return $(0, x)$; with probability $1/2$, set $x = \mathbb{A}$ and return $(1, x)$.
> > > > This distribution exhibits pseudo-entropy, as evidenced by a distribution where the first bit is replaced with a uniformly random bit. The construction of a distribution with pseudo-entropy is a crucial step toward constructing pseudorandom generators.
> > > >
> > > > While this outline represents only a roadmap, it reflects our reasoning behind the belief that such a proof exists. Moreover, one could argue that the existence of a distribution with pseudo-entropy, in itself, constitutes an example of a cryptographic primitive.
> > > >
> > > > ---
> > > >
> > > > ### Representation of $f$
> > > >
> > > > We would like to refer the reviewer to our earlier response (Specification of the Interactive Protocol, item 2), where we agreed with your initial comment that Alice and Bob should agree on a representation of $f$. We believe this clarification adequately addresses the concern regarding the truth-table issue.

---

### Meta-Review · Area_Chair_3RcR · 2024-12-23

**Metareview:**

This paper derives a theoretical analysis of the relationship between backdoor-based watermarks and adversarial defenses. The authors show that all discriminative learning tasks can be categorized into one of three classes, which suggests there is an inherent security trade-off in ML applications.

Reviewers generally appreciated the technical depth of the paper and its clever use of existing techniques to derive their theoretical result. However, most reviewers also found the paper's analysis framework to be lacking in precision, leaving much of the derived theoretical result up to interpretation. The paper is also written in a way that is difficult to absorb for the general ML audience. As a result, practical implications of the paper are also unclear. After the rebuttal period, reviewers and AC discussed thoroughly and reached consensus that the paper's weaknesses currently outweigh its strengths. Thus, AC believes the paper is currently not ready for publication, but encourages the authors to taken into account reviewer suggestions and improve the paper's clarity before resubmitting to a future venue.

**Additional Comments On Reviewer Discussion:**

Reviewers and authors discussed weaknesses cited in the meta-review. Although the authors did clarify some technical aspects, reviewers believe a major revision is required to make the analysis framework precise enough for publication.

---

### Decision · Program_Chairs · 2025-01-22

Reject